# Deciphering the Regulatory Potential of Antioxidant and Electron-Shuttling Bioactive Compounds in Oolong Tea

**DOI:** 10.3390/biology14050487

**Published:** 2025-04-28

**Authors:** Regineil A. Ferrer, Bor-Yann Chen, Jon Patrick T. Garcia, Christine Joyce F. Rejano, Po-Wei Tsai, Chung-Chuan Hsueh, Lemmuel L. Tayo

**Affiliations:** 1School of Chemical, Biological, and Materials Engineering and Sciences, Mapúa University, Manila 1002, Philippines; raferrer@mymail.mapua.edu.ph (R.A.F.); jptgarcia@mymail.mapua.edu.ph (J.P.T.G.); cjfrejano@mymail.mapua.edu.ph (C.J.F.R.); 2School of Graduate Studies, Mapúa University, Manila 1002, Philippines; 3Department of Chemical and Materials Engineering, National I-lan University, I-lan 260, Taiwan; boryannchen@yahoo.com.tw (B.-Y.C.); niucmorg2021@gmail.com (C.-C.H.); 4Department of Food Science, National Taiwan Ocean University, Keelung 202, Taiwan; powei@mail.ntou.edu.tw; 5Department of Biology, School of Health Sciences, Mapúa University, Makati 1200, Philippines

**Keywords:** Oolong tea, breast cancer, in vitro, flavonoids, antioxidant, electron shuttles, MFC, CV, in silico, network pharmacology, molecular docking, molecular dynamics

## Abstract

This study explored how natural compounds in Oolong tea (OT) may help fight breast cancer. We tested different extraction methods and found some compounds with strong antioxidants and others that boosted electrical activity. Computational methods showed that certain compounds, especially luteolin, could regulate cancer-related proteins. These findings suggest OT may have cancer-fighting and energy-stimulating properties; however, further research is needed.

## 1. Introduction

From ancient traditions to modern wellness trends, the evergreen shrub *Camellia sinensis* is one of the most widely consumed non-alcoholic beverages in the world due to the abundance of bioactive secondary metabolites [1]. The application of diverse processing techniques and degrees of fermentation gives rise to different types of tea: green tea (GT, unfermented), OT (semi-fermented), and black tea (BT, fully fermented) [2]. GT retains a higher catechin content than OT and BT because polyphenol oxidase in the tea leaves catalyzes the oxidation of catechin into theaflavins during the fermentation process [3]. Chen et al. further explored the electrochemical activity of various *Camellia* tea extracts, revealing that their redox potential curves were associated with the polyphenolic content, ranking GT as the most electrochemically active, followed by OT and BT [4]. This sets the stage for investigating OT specifically, highlighting its potential in disease treatment. Partially fermented OT has been valued not only for its distinct flavor and aroma but also for its numerous health benefits. Extensive research has explored the abundance of bioactive phytochemical constituents in OT, such as polyphenols, flavonoids, phenolic acids, condensed tannins, glycosidic acids, and volatile oils [5,6,7,8,9]. Much interest has also been attracted to its investigation, as these secondary metabolites exhibit broad pharmacological effects, including antioxidant, anti-inflammatory, antimicrobial, hypoglycemic, and chemopreventive properties [10,11]. Among these, compounds containing phenyl groups and conjugated systems for redox-mediating bioconversion and biotransformation are linked with antioxidative and anticancer properties by reducing free radicals and preventing cellular damage [12,13]. Furthermore, certain compounds have shown electron-shuttling capabilities due to their electrochemical stability, reversibility, and efficiency in transferring electrons to and from electron donors and acceptors. Electron shuttles (ESs) are characterized by the presence of dihydroxyl constituents in the *ortho*- and *para*-positions in the benzene ring of these electron-shuttling compounds that are significantly contributing to their pharmacological effects on the human body, suggesting their potential as substitutes for commercially available artificial drugs [14,15].

The anticancer potential of tea polyphenols, particularly those derived from *C. sinensis*, has been extensively studied, with growing evidence highlighting their role in cancer prevention and treatment. Among various malignancies, BC remains one of the most prevalent and deadliest, accounting for 2.3 million cases and 670,000 deaths in 2022 [16,17]. While BC often occurs sporadically, genetic mutations in *BRCA1* and *BRCA2* have been found to be significant in its development. Moreover, the incidence of such alterations is aggravated by an unhealthy lifestyle, which includes alcohol abuse, smoking, and poor dietary habits [18,19]. The molecular heterogeneity of BC is categorized into four major subtypes: luminal A (LaBC), luminal B (LbBC), HER2-positive (Her2BC), and triple-negative (TNBC). These classifications are defined by the variation in expression patterns of key hormone receptors estrogen (ER), progesterone (PR), and human epidermal growth factor receptor 2 (HER2) [20]. The presence of ER and/or PR, absence of HER2, and low Ki-67 expression levels are common manifestations of the LaBC subtype, owning favorable survival rates and prognosis. Conversely, the LbBC subtype exhibits the presence of ER and high expression levels of Ki-67, while PR may be less likely observed, with a higher histologic grade and more aggressive profile than LaBC. On the other hand, the Her2BC subtype has high expression levels of HER2 yet lacks both ER and PR, which develops more rapidly and aggressively than the luminal subtypes. Lastly, the TNBC subtype shows the absence of ER, PR, and HER2, as indicated in its name. In addition to being a common subtype among women under 40 years old, TNBC is characterized by an early relapse and a high probability of presenting during advanced stages of the disease. Gene expression analysis through DNA microarray technology has become an important tool for investigating the dysregulated genes of BC compared to normal cells. The use of such an approach enabled the systematic selection of hub gene targets, identifying crucial genes that serve as protein targets of the OT bioactive compounds. Among the collection of genes, the aberrant nature of *PIK3CA* and *MAPK1* in luminal and HER2 BCs is the key player in the oncogenicity of BC development [21,22], underscoring the potential of OT as a therapeutic agent for targeted cancer treatment.

Current BC treatment strategies, including hormone therapy, chemotherapy, and surgery, have shown promising clinical outcomes but come with significant drawbacks, such as invasive procedures, expensive medications, and serious side effects. With the intention of addressing these concerns, natural products are being explored for their potential anticancer properties and continue to be a subject of interest to optimize their therapeutic claims. Several studies have demonstrated the anti-tumor effects of tea extracts in various cancer models. For instance, Wang et al. made use of A/J mice models to demonstrate the inhibitory mechanisms of tea against lung tumorigenesis [23]. They found a significant reduction in tumors when administering decaffeinated green or black tea in mice during or after NNK treatment. The same conclusions were reported in the study of Yang et al. when they observed reduced progression of adenomas to adenocarcinomas among mice that had been given black tea 16 weeks prior to NNK injection [24]. On the other hand, Hajiaghaalipour et al. utilized white tea extracts and evaluated their inhibitory effects on HT-29, which is a colorectal cancer cell line [25]. Results showed that at a concentration of 87 μg/mL, the proliferation of the cancer cells is obstructed. They also emphasized that aside from the antioxidative and antiproliferative properties of white tea, it prevents DNA damage among normal cells as well. Meanwhile, black tea extracts were reported to inhibit the growth of SW1990 PANC-1 human pancreatic cancer cells and SW116 human colorectal cells by decreasing AKT activity and increasing p38 activity [26]. Consequently, several articles have supported the use of different *C. sinensis* extracts as treatment specifically for BC progression. Using in vitro assays and in vivo mouse models, Sartippour et al. found that catechin components from green tea extracts were effective inhibitors against BC cell proliferation [27]. Moreover, TF-1, TF-2, and TF-3, which are black tea polyphenols, showed antiproliferative effects on hormone-resistant BC cells after suppression of basal receptor tyrosine phosphorylation in HER2/neu-expressing MCF-7 cells was observed [28]. In fact, these compounds were explored in silico to further understand their underlying molecular mechanisms in ameliorating BC development. Saini et al. investigated the interactions of polyphenols—gallic acid, quercetin, resveratrol, piperine, and beta-sitosterol—with key mammary BC hub genes (*ESR1*, *CYP19A1*, and *EGFR*) identified via network pharmacology [29]. Their findings revealed that these compounds exhibited better binding affinities than positive control drugs (raloxifene core, letrozole, and erlotinib) to the target proteins. Molecular dynamics confirmed the stability of these interactions, highlighting key amino acid interactions that may contribute to their therapeutic potential. Similarly, in vitro studies on epigallocatechin-3-gallate (EGCG) demonstrated its ability to reduce the enzymatic activity of tyrosine phosphatase PTP1B and decrease the viability of MCF-7 cells [30]. Molecular simulation assays supported these findings, revealing that strong ligand binding stabilized the complex conformation in line with its inhibitory properties. Catechin derivatives were investigated for their role in TNBC tumorigenesis using in vitro and in silico approaches, revealing interactions with critical hub genes (*CASP3*, *MAPK14*, *PPARG*, *MMP3*, *SERPINE1*, *SRC*, *BCL2*, *EGFR*, *MMP9*, and *KDR*) that contribute to BC mitigation through DNA damage repair from UV-A exposure and lipid peroxidation-induced apoptosis, while also exhibiting strong binding affinity to extracellular domains of EGFR, comparable to the control drug erlotinib [31].

Although many studies have been focused on elucidating the anticancer potential of various *C. sinensis* extracts, such as green tea and black tea, there is limited research on the effects of polyphenol compounds specifically from OT against BC. In particular, considering electrochemical perspectives (e.g., electron-shuttling characteristics) to decipher the MOA of disease-treating efficacy has remained open to exploring wide-ranging medicinal applications. Despite this insufficiency, these studies have proven that OT does have anticancer properties that are effective against BC. One is the study of Shi et al., wherein they reported the capacity of OT to induce signaling pathways that result in the damage and cleavage of DNA, leading to the inhibition of BC tumorigenesis [32]. The bioactivity of herbal tea compounds has been linked to their electrochemical properties, with MFC and CV used to assess their electron-shuttling capabilities. Thus, this study was intended to provide further evidence on the mechanistic action of polyphenolic compounds from OT and their potential regulatory effects on BC progression. Through in silico approaches, we specifically aimed to identify the deregulated risk genes found in different BC subtypes, LaBC, LbBC, Her2BC, and TNBC, that are targeted by electron-shuttling compounds from OT and investigated their combined interactions through network pharmacology and protein–ligand bioinformatics, i.e., molecular docking and dynamics simulations. For in vitro approaches, we also compared different extraction methods, such as SE and SFE, in two solvent systems (water and ethanol) to determine which technique would be the most optimal for obtaining these compounds by analyzing their phytochemical content, antioxidant activity, and electrochemical properties.

## 2. Materials and Methods

### 2.1. Sample Preparation

OT bags were purchased from a local supermarket in Tainan City, Taiwan. The removal of essential oils from the OT bags was performed using the Supercritical Fluid Preparative Scale CO_2_ Extraction Equipment (SE-2000-10000 Series; Taiwan Supercritical Technology Co., Ltd.; Changhua, Taiwan). The extraction process involved placing 200 g of OT bags in the extraction chamber. The system was set to a temperature of 35 °C and a pressure of 2000 psi, and the extraction was conducted for 60 min. The essential oils were successfully removed from the OT bags during this procedure. Following this removal of oil, OT was extracted using two different solvents, namely, water and ethanol, in a 1:20 (herb:water) ratio. The samples extracted using ethanol were refluxed at 60 °C for 2 h. In contrast, the other samples subjected to water extraction were boiled in a traditional Chinese decoction pot until the volume was reduced to approximately 200 mL. The extracts were filtered through vacuum filtration and were concentrated using a rotary evaporator. Once concentrated, the samples were transferred into 50 mL centrifuge tubes and stored in the freezer for 24 h. Then, the frozen crude samples were subjected to a 3-day lyophilization process using a freeze dryer.

### 2.2. Phytochemical Content

#### 2.2.1. Total Polyphenol Content

To determine the total polyphenol content (TPC) present in OT, the same procedure described by Tsai et al. was carried out with slight modifications to optimize the results [33]. Ten (10) mg of gallic acid was dissolved in 10 mL of ethanol to have a 1 mg/mL standard stock solution. Then, the stock solution was used to dilute gallic acid by a factor of two to have six (6) varying concentrations (e.g., 1000, 500, 250, 125, 62.5, 31.3 μg/mL). For the sample solution, the freeze-dried sample was dissolved using the appropriate solvent (water and ethanol for water and ethanol extracted OT) to obtain a 10 mg/mL stock solution. All the prepared solutions were treated with 500 μL Folin–Ciocalteu reagent and 400 μL Na_2_CO_3_ solution. After incubating the samples for 30 min, 200 μL of each test solution was placed in triplicate in a 96-well plate. The absorbance was measured at 600 nm using an ELISA microplate reader (Multiskan skyhigh microplate Spectrophotometer (A51119700DPC); Thermo Fisher Scientific; Taipei, Taiwan).

#### 2.2.2. Total Flavonoid Content

The method for determining total flavonoid content (TFC) was adapted from Lee et al. [34]. One (1) mg/mL of rutin in ethanol was prepared as the standard stock solution for the determination of total flavonoid content. Six different concentrations (e.g., 1000, 500, 250, 125, 62.5, 31.3 μg/mL) were obtained from the stock solution via two-fold serial dilution. Meanwhile, 1.25 mg/mL of sample solution was prepared for analysis. Then, 500 μL of each stock solution, including blanks, was treated with 500 μL of 2% AlCl_3_. After 15 min of incubation, 200 μL of each test solution was added to a 96-well plate, and the absorbance was measured at 430 nm using an ELISA microplate reader.

#### 2.2.3. Total Condensed Tannin Content

For total condensed tannin content (TCTC) analysis, catechin standard solution was prepared in various concentrations via two-fold serial dilution starting from 1 mg/mL to 0.0313 mg/mL. Whereas the sample extract was prepared in 1.25 mg/mL solution. Then, 600 μL of vanillin in H_2_SO_4_ was added to 300 μL of each stock solution. After 15 min, 200 μL of each test solution produced in triplicate was transferred to a 96-well plate. The absorbance was measured at 530 nm using an ELISA microplate reader.

### 2.3. Antioxidant Activity

#### 2.3.1. 2,2-Diphenyl-1-picrylhydrazil Free Radical Scavenging Capacity

A standard stock solution was prepared by dissolving 10 mg of ascorbic acid in 10 mL ethanol. Then, different concentrations (e.g., 1000, 500, 250, 125, 62.5, 31.3, 15.63, 7.81 μg/mL) were obtained from serially diluting the 1 mg/mL stock solution by a factor of two. 1 mg/mL of sample stock solution was also prepared and serially diluted by a factor of two, which resulted in different concentrations. Briefly, 50 μL of each sample extract, standard solution, and blanks were treated with 150 μL of 2,2-diphenyl-1-picrylhydrazil (DPPH) solution and reacted for 30 min in the dark. After incubation, the absorbance was measured using an ELISA microplate reader at 517 nm. All samples were produced in triplicate to ensure the reliability of the data. For analysis, Equation (1) was used for the calculation of the radical scavenging activity (%RSA), while linear regression was utilized to estimate the half-maximal inhibitory concentration (IC_50_) at 50% radical scavenging activity.(1)$$\%\;RSA=\frac{\left(A_{ctl}-A_{blk}\right)-\left(A_{spl}-A_{blk}\right)}{\left(A_{ctl}-A_{blk}\right)}\times 100$$
where % *RSA*, *A_ctl_*, *A_blk_*, and *A_spl_* represent the percentage of radical scavenging activity, the absorbance of the control sample, the absorbance of the blank, and the absorbance of the sample, respectively.

#### 2.3.2. Ferric Reducing Antioxidant Power Assay

Five hundred (500) μg/mL of Trolox stock solution was diluted with 500 μL of a 2:3 ratio of ethanol and D.D. water to obtain a 1000 μg/mL standard stock solution. A series of standard stock solutions was prepared via two-fold serial dilution to have the following concentrations: 1000, 500, 250, 125, 62.50, 31.25, and 15.63 μg/mL. One mg/mL of sample stock solution was prepared. Then, 50 μL of each sample extract and standard solutions were transferred to a 2000 μL microcentrifuge tube and treated with 1450 μL of ferric reducing antioxidant power (FRAP) reagent. After treatment, triplicates of 200 μL of each test solution were placed on a 96-well plate. The absorbance was measured at 593 nm using an ELISA microplate reader. Blanks of each sample were also prepared and measured to determine the limit of quantification.

### 2.4. Electrochemical Property

#### 2.4.1. Power Density Through Double Chambered-Microbial Fuel Cell

The model double-chambered MFC set-up used for power-generating testing comprised a cathodic and an anodic chamber separated by a proton exchange membrane (DuPont™ Nafion^®^ NR-212; Tyrone, PA, USA) with a contact area of ca. 4.52 cm^2^ (ID = 1.2 cm). The graphite cathode and anode (Grade: IGS743; Central Carbon Co., Ltd.; Taipei, Taiwan), soaked in the electrolyte and culture broth during DC-MFC operations, have a contact area of ca. 1.649 cm^2^. The cathodic chamber contained the electrolyte solution composed of 17.429 g K_2_HPO_4_ (dipotassium hydrogen phosphate; BAKER ANALYZED, A.C.S. Reagent) and 6.371 g K_3_Fe(CN)_6_ (potassium ferricyanide; Sigma Aldrich; St. Louis, MO, USA) dissolved in 200 mL deionized-distilled (DD) water. On the other hand, culture broth containing *Aeromonas hydrophila* NIU01 electroactive bacteria was added to the anodic chamber. This electroactive bacterium is a nanowire-generating microbe capable of bioelectricity generation and was originally isolated and enriched through the serial selection of acclimated populations of azo dye decolorizer. To have consistent bacterial cultures, 100 mL of autoclaved LB broth culture medium (Difco LB Broth, Miller; Luria-Bertani; 10 g/L tryptone, 5 g/L yeast extract, 10 g/L NaCl) was prepared and seeded with *A. hydrophila* NIU01. Incubation was carried out using a water bath shaker (SKW-12; Shin Kwang Machinery Industry Co. Ltd.; Taipei, Taiwan) for 12 h at 30 °C, 125 rpm. Subsequently, 1% (*v*/*v*) of the seeded broth was added to a new set of sterilized LB broth, then cultured for 12 h. Here, the pH level of the flask cultures was not controlled during the bacterial batch culture.

Furthermore, quantitative analysis was performed by adding various sample concentrations to the anode component (250, 500, 750, 1000, 1500, and 2000 ppm, respectively). Stock solution A (1.2 g of OT extract in 12 mL DD water for water extract (or 11 mL DD water + 1 mL ethanol for ethanol extract samples)) was prepared to achieve the sample concentrations in a constant working volume (200 mL) of the anode compartment. After each concentration, the anode chamber and electrode were rinsed twice with DD water. For the consistency of the results, the following practices were adhered to: (1) optical density (OD_600_) of 2.1 was maintained, and (2) blank 1 before 250 ppm and (3) blank 2 after 2000 ppm were conducted. For the standard and control, 0.3 mM of dopamine was employed after blank 2 by adding 0.1 mL of 0.6 M to the fresh anodic chamber. Time courses of voltage and electric current were calculated using Equations (2) and (3) with a D/A system. External resistance of 1.0 kΩ was applied to MFC for the basis of comparison.(2)$$P_{density}=\frac{{(V}_{MFC})(I_{MFC})}{A_{anode}}$$(3)$$I_{density}=\frac{I_{MFC}}{A_{anode}}$$
where the parameters *V_MFC_* and *I_MFC_* were directly measured from the workstation by applying the linear sweep voltammetry option for electrochemical analysis (ZIVE SP1; Won-A-Tech Co. Ltd., Bucheon, Republic of Korea). The *A_anode_* was denoted as the working area of the graphite anodes used during the power density (PD) measurement.

#### 2.4.2. Cyclic Voltammetry Assay

To comparatively investigate the electrochemical characteristics of the *C. sinensis*, 50-cycle CV scanning on an extract-bearing solution (ca. 1000 ppm) was conducted using the ZIVE SP1 electrochemical workstation (Won-A-Tech Co. Ltd., Republic of Korea) with a voltage range set from −1.5 V to +1.5 V and a scan rate of 10 mV/s through the electrochemical workstation. A three-electrode system was employed in the setup. The working electrode used herein was a glassy carbon electrode (GCE, ID = 3 mm; model CHI104; CH Instruments Inc.; Bee Cave, TX, USA) with a contact area of 0.07 cm^2^ and was polished with 0.05 µm alumina polish (aluminum oxide, <50 nm particle size; Sigma Aldrich) and subsequently rinsed with deionized water before use to prevent interference. The GCE was modified with 10 µL of carbon black solution and 10 mg of carbon black dissolved in 1 mL of dimethylformamide (DMF), placed on the surface of the base of the disk electrode, and left at room temperature to air dry for 12 h. For the reference electrode, a Hg/Hg_2_Cl_2_ electrode filled with saturated aqueous potassium chloride (KCl) was used. Finally, the counter electrode used to conduct CV experiments was a platinum electrode with a working area of ca. 6.08 cm^2^. Prior to the actual measurement, the samples were purged with gaseous nitrogen for 20 min to ensure the removal of residual oxygen. For one cycle, a symmetric scan ranging from − 1.5 V to + 1.5 V was applied to prevent water electrolysis from occurring during CV analysis. As for the scan rate, a value of 100 mV/s was used for the experiment. Smart manager software v6.6 was used to evaluate the closed-loop areas of the redox potentials in CV curves as an indicator of electrochemical potentials. The areas enclosed by the redox potential curves in CV loops (see Equation (4)) were described as indicators of the electrochemical characteristics of the extract-bearing solutions using SigmaPlot 10.0. For the calculations, V_H_ and V_L_ were reported as the maximum and minimum CV scanning voltages (i.e., +1.5 and −1.5 V) used for the run, respectively, whereas i_h_ and i_l_ represented the oxidation and reduction currents at a given scan voltage, respectively.(4)$$Area=\int_{V_{L}}^{V_{H}}(i_{h}-i_{l})dV$$

### 2.5. Network Pharmacology

#### 2.5.1. Screening and Evaluation of Oolong Tea Bioactive Compounds

The bioactive compounds present in OT were sourced from previous literature [5,6,7,8,9] and from Indian Medicinal Plants, Phytochemistry, and Therapeutics (IMPPAT) v2.0 [https://cb.imsc.res.in/imppat/ (accessed on 8 August 2024)] and Phytochemical Interactions Database (PCIDB) [https://www.genome.jp/db/pcidb (accessed on 8 August 2024)]. IMPPAT is a curated database that integrates the phytochemical contents of traditional therapeutic Indian medicine and their absorption, distribution, metabolism, excretion, and toxicological properties [35]. Meanwhile, PCIDB incorporates data from various bioinformatics repositories, i.e., KNApSAcK, ChEMBL, and CTD, to provide insights into plant metabolite–protein interactions and their implications on human health [36]. The SMILES of the compounds were obtained from PubChem [https://pubchem.ncbi.nlm.nih.gov/ (accessed on 13 August 2024)] [37], and these were submitted to SwissADME [http://www.swissadme.ch/ (accessed on 15 August 2024)] and ADMETLab v3.0 [https://admetlab3.scbdd.com (accessed on 15 August 2024)] to screen and evaluate their pharmacokinetic potential based on their physicochemical properties [38,39]. In SwissADME, compounds with a high gastrointestinal absorption score and a bioavailability score of greater than or equal to 0.55 were included, while in ADMETlab, compounds with a human intestinal absorption score of less than 0.3 were considered.

Lipinski’s and Veber’s rules were also taken into account in both screening tools. Lipinski’s rule of five assesses drug likeness by ensuring the suitability of a compound for oral administration [40]. The drug likeness of a compound is assessed by the following conditions: (i) the number of hydrogen bond donors must be less than 5, (ii) the number of hydrogen bond acceptors must be less than 10, (iii) the molecular weight must be less than 500 Da, and (iv) the logarithmic partition coefficient must be less than 5. In contrast, Veber’s rule assesses the bioavailability of a compound through the following conditions: (i) the number of rotatable bonds must be fewer than 10, and (ii) the total polar surface area must be less than 140 Å [41].

Then, the chemical structure of the compounds that passed the screening tests was further evaluated to determine which of these possess electron-shuttling properties, as previously described by Hsueh et al. [15], focusing on phenolic compounds with *ortho*- and *para*-dihydroxyl substituents.

#### 2.5.2. Identification of Target Gene Candidates

Target genes of the previously screened and evaluated OT bioactive compounds were then determined using two machine-learning programs: SwissTargetPrediction [http://www.swisstargetprediction.ch (accessed on 20 August 2024)] and SuperPred [https://prediction.charite.de (accessed on 20 August 2024)]. SwissTargetPrediction is a ligand-based target prediction tool that utilizes the Tanimoto index and Manhattan distance similarity for two-dimensional and three-dimensional similarity measures, respectively, trained by multiple logistic regression [42]. SuperPred, similarly, is used for the target prediction of small compounds and utilizes the anatomical therapeutic chemical (ATC) classification system for drug cataloging [43]. Only those genes that have at least 10% probability in SwissTargetPrediction and 60% probability with a model accuracy of 90% in SuperPred were considered qualified for further downstream analyses.

#### 2.5.3. Differential Expression Analysis of Breast Cancer Subtypes

Microarray data of the four BC subtypes were retrieved from the National Center for Biotechnology Information—Gene Expression Omnibus (NCBI-GEO) [https://www.ncbi.nlm.nih.gov/geo/ (accessed on 3 October 2024)]. NCBI-GEO is an open-access repository of genomic data, comprising different high-throughput sequencing datasets from various research institutions [44]. The chosen dataset (GSE45827) for this study is comprised of human genome expression profiles, analyzed using the Affymetrix U133 Plus 2.0 Array. It contains 29 LaBC samples, 30 LbBC samples, 30 Her2BC samples, and 41 TNBC samples, all of which were obtained from BC tumor specimens, which were compared to 11 normal tissue samples.

The method for differentially expressed gene (DEG) analysis was adapted from the paper of Garcia and Tayo [45]. Differential expression analysis using GEO2R [https://www.ncbi.nlm.nih.gov/geo/geo2r (accessed on 3 October 2024)] was conducted to obtain DEGs from each of the four BC subtypes relative to the controls. GEO2R is a bioinformatics tool utilized to compare two or more groups of sequencing data to determine DEGs across varying experimental conditions. The following parameters were set: (i) the Benjamini and Hochberg false-discovery rate for *p*-value adjustment to reduce false positives [46], (ii) limma precision weights for accounting for the mean–variance relationship [47], and (iii) force normalization for log transformation and consistent value distribution across samples. Those genes with an adjusted *p*-value of <0.05 were retained and grouped either as downregulated (FC < −1.5) or upregulated (FC > 1.5). Consequently, the DEGs identified from each subtype were matched with the target gene candidates of the bioactive compounds from OT using an online Venn diagram tool [https://bioinformatics.psb.ugent.be/webtools/Venn/ (accessed on 5 October 2024)] to determine the genes from the four BC subtypes targeted by these compounds.

#### 2.5.4. Construction of Protein–Protein Interaction Networks for Identifying Hub Genes

Cytoscape v3.10.1 [https://cytoscape.org/index.html (downloaded on 7 October 2024)] [48] was used to create the protein–protein interaction networks between common target genes between the OT and different BC subtypes identified using Venny 2.1. Cytoscape plugins, such as stringApp v2.1.1 https://apps.cytoscape.org/apps/stringapp (accessed on 5 October 2024)] and cytoHubba v0.1 [https://apps.cytoscape.org/apps/cytohubba (accessed on 5 October 2024)], were utilized in the process. The Search Tool for Retrieval of Interacting Genes (STRING), a database of known and predicted protein–protein interactions sourced from genomic context predictions, high-throughput lab experiments, co-expression, and automated text-mining [49], was used to create the protein–protein interaction network related to “*Homo sapiens*”. A 70% confidence cutoff score for interaction significance, disregarding gene interactions with no more than 2 edges. Topological analysis of the constructed pharmacological network was performed using cytoHubba by determining its topological algorithms, specifically maximal neighborhood component (MNC), maximal clique centrality (MCC), degree, and closeness. The top 15 hub genes were retrieved from each algorithm, and the overlapping hub genes in all three algorithms were identified using a Venn diagram for further downstream analyses [50].

#### 2.5.5. Gene Ontology Analysis

The underlying biological mechanism of key target genes of OT related to BC subtypes was submitted and analyzed through enrichment analysis using Database for Annotation, Visualization, and Integrated Discovery [DAVID, https://david.ncifcrf.gov/home.jsp (accessed on 21 December 2024)]. DAVID is a bioinformatics tool that provides functional annotation programs for understanding the biological mechanisms of genes, including their functions and pathways [51,52]. In this study, the gene ontology (GO) database was used to describe and define genes based on their biological processes (BP), cellular components (CC), and molecular functions (MF). Terms with *p*-values < 0.05 were considered statistically significant in the analysis. The top 20 terms, ranked by descending -log10(*p*-value), were visualized using the Scientific and Research Plot [SRplot, https://www.bioinformatics.com.cn/srplot (accessed on 21 December 2024)] [53].

### 2.6. Molecular Simulations

#### 2.6.1. Molecular Docking

The electron-shuttling OT compounds with available 3D conformational structures from PubChem were docked to the proteins associated with their hub genes identified through network pharmacology results. The 3D conformation structures of selected compounds were obtained from PubChem in SDF format and converted to mol2 format using OpenBabel v3.1.1 [https://github.com/openbabel/openbabel/releases (accessed on 9 December 2024)] [54]. OpenBabel is a C++-powered program used for interconverting molecular file formats used in modeling and computational chemistry. Hydrogen addition and geometry optimization of the ligands were performed using Avogadro v1.2.0 [https://avogadro.cc/ (accessed on 9 December 2024)] [55]. Crystal structures of proteins associated with the hub genes were sourced from the Research Collaboratory for Structural Bioinformatics Protein Data Bank [RCSB PDB, https://www.rcsb.org/ (accessed on 9 December 2024)] in PDB format [56]. Here, the selection process of the PDB IDs of the proteins must have their crystal structures determined using X-ray crystallography with resolutions ranging from 1.80 to 2.20 Å [57,58,59]. Proteins of great resolution were then subjected to P2Rank: PrankWeb server [https://prankweb.cz/ (accessed on 9 December 2024)] to determine the presence of binding sites in the protein (see Table 1), meeting the following criteria [60,61]: (1) a probability score of greater than 90%, (2) solvent-accessible surface points of greater than 100, and (3) a high P2rank score relative to other pocket active sites. Heteroatoms, ions, and water molecules were removed from the protein crystal structures using Biovia Discovery Studio 2024 v24.1.0.23298 [https://discover.3ds.com/discovery-studio-visualizer-download (downloaded on 9 December 2024)] [62]. Subsequently, missing atoms, polar hydrogens, and Autodock4 atom types were added to the protein using AutoDock Tools v1.5.7 [https://vina.scripps.edu/ (downloaded on 9 December 2024)] [63], with structures exported in PDBQT format. Gasteiger charges were applied to the ligands, and all flexible bonds were set as rotatable. Docking analysis was performed using Auto-Dock Vina v1.1.2 [https://vina.scripps.edu/ (downloaded on 9 December 2024)] [64,65], an optimized molecular docking program based on the AutoDock4 scoring function. Vina utilizes an empirical scoring function and a hybrid global–local search algorithm, combining a Monte Carlo iterated search with a BFGS gradient-based optimizer to predict ligand binding affinity in kcal/mol [65,66].

The interactions of the amino acids in the proteins with the ligands were investigated using Biovia Discovery Studio Visualizer v24.1.0.23298. The lowest conformational position of the ligands and the protein under scrutiny was imported into the software. Only favorable non-bond interactions, specifically hydrogen bonds and electrostatic interactions, were selected for the analysis from 2D and 3D perspectives.

#### 2.6.2. Molecular Dynamics

Molecular dynamics simulations were performed using GROMACS 2024 [https://manual.gromacs.org/current/download.html (downloaded on 15 November 2024)] [73] to evaluate the interaction between electron-shuttling ligands and key hub proteins. The CHARMM36 all-atom force field was employed to parameterize the systems. For ligand topology optimization, the ligand from the protein–ligand complex was extracted, which was then submitted to Avogadro for hydrogen addition to ensure proper protonation and exported as a mol2 file format. The bond orders were rearranged in ascending order, and the ligand name was standardized. The MOL2 file was submitted to CGenFF v4.0 [https://cgenff.com/ (accessed on 17 November 2024) [74] for CHARMM General Force Field (CHARMM-GFF) parameterization. This was then converted to a GROMACS-compatible format with the CHARMM36 force field for molecular dynamics simulations. All scripts used for ligand topology were obtained from https://manual.gromacs.org/2024.4/index.html (accessed on 9 November 2024), http://www.mdtutorials.com/gmx/complex/02_topology.html (accessed on 9 November 2024), and https://mackerell.umaryland.edu/charmm_ff.shtml (accessed on 9 November 2024). Each docked system was positioned within a cubic simulation box, maintaining a 1.0 nm distance from the box boundaries, and solvated with water using the CHARMM-modified TIP3P water model. To ensure system neutrality, Na^+^ or Cl^−^ counterions were added in proportion to the system’s net charge. Energy minimization was conducted using the steepest descent integrator for 50,000 steps with a convergence threshold of 1000 kJ/mol. This was followed by a 100 ps NVT equilibration phase at 300 K using a V-rescale thermostat, and a 100 ps NPT equilibration phase at 300 K and 1 bar employing a C-rescale barostat. Subsequently, a 200 ns production run was executed using the leap-frog integrator, with pressure maintained by the Parrinello–Rahman barostat and a time constant of 2 ps. Simulation data, including trajectories and energies, were recorded at 10 ps intervals. Throughout the simulations, long-range electrostatics were calculated using the particle mesh Ewald (PME) method, with a 1.0 nm cutoff applied for both electrostatic and van der Waals interactions. Post-simulation analyses included the calculation of root mean square deviation (RMSD), root mean square fluctuation (RMSF), total radius of gyration (Rg), solvent-accessible surface area (SASA), and hydrogen bond formations, providing detailed insights into protein–ligand interactions.

#### 2.6.3. Molecular Mechanics Poisson–Boltzmann Surface Area

To validate the molecular dynamics results, molecular mechanics Poisson–Boltzmann surface area (MM/PBSA) was performed to calculate the binding free energy of the ligand-bound hub proteins with stable trajectory-based data [75]. The system was modeled using the Amber force field with the leaprc.ff99SBildn and leaprc.gaff parameter sets for protein and ligand, respectively. The analysis was performed using a stable trajectory from 100 to 120 ns. Poisson–Boltzmann (PB) calculations were carried out with an internal dielectric constant of 1.0, an external dielectric constant of 80.0, and an ionic strength of 0.15 M. A probe radius of 1.4 Å was used to define the solvent accessible surface, and the solvent surface tension was set to 0.0378 kcal/mol·Å^2^. For residue-level decomposition, a decomposition method of 2 was chosen, with energy contributions printed within a 4-angstrom cutoff radius.

#### 2.6.4. Principal Component Analysis/Free Energy Landscape

Principal Component Analysis (PCA) and Free Energy Landscape (FEL) analysis were performed to investigate the conformational dynamics of the ligand-bound hub proteins [76]. The covariance matrix was computed using gmx covar on the 0–200 ns trajectory with analysis conducted on the first, stable, and last 20 ns window frames [73]. For each time window, eigenvalues and eigenvectors were calculated, followed by PCA using the first two principal components with gmx anaeig to capture the essential dynamics [73]. The free–energy landscape was generated using gmx sham, with energy levels plotted based on the principal components.

## 3. Results

### 3.1. Phytochemical Analysis

Since some of the polyphenols, flavonoids, and condensed tannins are aromatics-bearing *ortho*- or *para*-dihydroxy substituents, these phytochemical contents were explored for their association with the pharmacological impact of OT. The phytochemical analyses of OT extracts processed and extracted in different methods are presented in Table 2. By comparison, the TPC of the OT extracts was calculated from the regression equation of the calibration curve (y = 3.8354x − 0.0132, R^2^ = 0.9998), expressed in gallic acid equivalents (GAE). Furthermore, the TFC of OT extracts derived using the calibration curve (y = 4.9008x + 0.0287, R^2^ = 0.9994) was expressed in rutin equivalents (RE). Likewise, the calibration curve from linear regression (y = 7.7867x − 0.0242, R^2^ = 0.9974) was used for the TCTC analysis, which was expressed in catechin equivalents (CE). All the phytochemical content assays revealed that the OT-E extract possesses the highest concentration of polyphenols, flavonoids, and condensed tannins among all the extracts.

### 3.2. Antioxidant Analysis

Plants, including OT as herbal tea, naturally biosynthesize non-enzymatic antioxidants to attenuate oxidative damage caused by the reactive oxygen species (ROS) [77,78]. These polyphenol-enriched teas scavenge ROS through hydrogen atom transfer (HAT) and single electron transfer (SET) [79], which significantly contribute to their pharmacological impact. Since there are myriads of chemical species bearing free-radical scavenging capabilities at different levels and therefore no single antioxidant assay could comprehensively determine antioxidant activity, two bioassays were employed to unravel the antioxidant properties of the polyphenols and flavonoids in OT, as presented in Table 3. The DPPH radical scavenging assay was used to assess the ability of antioxidants to stabilize DPPH radicals via HAT. In this assay, the IC_50_ was used to evaluate the efficacy of the extract, indicating the concentration required to reduce the DPPH radical by half [80]. A lower IC_50_ value implies greater antioxidant activity. Meanwhile, the FRAP assay was used to determine the capability of OT antioxidants to reduce the Fe^3+^/tripyridyltriazine complexes to Fe^2+^/tripyridyltriazine via SAT [81,82]. Such reduction implies a direct relationship between the Trolox equivalents in the FRAP assay and antioxidant activity.

### 3.3. Bioenergy and Electrochemical Assessment via Microbial Fuel Cell and Cyclic Voltammetry

The electron-shuttling properties of bioactive compounds through bioelectricity generation have been shown to exert physiological effects that may aid in alleviating various diseases, such as Parkinson’s disease, viral infections, and neural degeneration [14,83,84,85]. Figure 1 and Table 4 illustrate the assessment of bioenergy-stimulating characteristics of ESs present in OT in different extraction methods using DC-MFC, with dopamine as a standard. All MFC profiles showed a direct correlation between the tea extract concentrations and power density generation, except for the OT-E extract, which exhibits an inverse relationship. Among various extraction methods for OT, the OT-W at 2000 ppm demonstrated the highest power density (PD) amplification. Taking the concentration 750 ppm as a point of comparison, OTL-W still exhibited the highest PD amplification, followed by OTS-E, OTL-E, and OTS-W. Regardless of the fact that the OTL-E has a decreasing trend, its power density was amplified even at its lowest concentration of OTL-E (250 ppm, 2.10 ± 0.57), greater than the 750 ppm of OTL-W (2.04 ± 0.25) if compared to their respective blanks.

Serial cycles of CV were performed to evaluate the electrochemically active OT compounds. The OT extracts of varying extraction methods and solvent systems were subjected to repeated cycles of reduction and oxidation potentials to investigate their electron-shuttling and antioxidant characteristics. Its closed-loop CV curve area serves as a measure for determining the electrochemical activity in terms of stability and reversibility. As presented in Figure 2 and Appendix A, regardless of pH conditions, all OT water extracts showed favorable electrochemical activities as catalysts, showing peaks in both oxidative and reductive regions in the voltammograms. In contrast, the OT ethanol extracts revealed antioxidant activity with significant oxidative peaks. The CV area of these electrochemical profiles was attenuated, as exhibited in Figure 3. After 50 cycles of CV scans, it was found that the area under the curve (AUC) of OTL-W extracts was greater than that of OTS-W extracts, and eventually, all of these water extracts steeply decreased. The stabilization of the CV area of OTL-W pH 9.50, OTS-W pH 8.18, and OTS-W pH 9.50 extracts was found to occur after 30 cycles until the 50th cycle. On the other hand, all OT ethanol extracts remained relatively stable throughout the 50-cycle CV run. A slow electrochemical degradation occurred in the AUC of OTS-E pH 9.50, while an abrupt decrease in CV areas of OTL-E pH 8.30 and OTS-E pH 8.41 was observed and stabilized after 15 cycles of CV scan.

### 3.4. Network Pharmacology

Initially, 265 compounds from OT were identified through literature and databases. These compounds were filtered based on their bioavailability, bioabsorption, drug likeness, and electron-shuttling properties, shown in Appendix A, yielding 22 compounds as candidates for drug discovery across various BC subtypes. The gene targets of these OT compounds were determined using machine learning-based target gene prediction tools. Additionally, the deregulated genes present in BC subtypes were obtained from the microarray dataset GSE45827. A notable observation was the discrepancy in the LaBC gene dataset, where *PIK3CA* appeared as both upregulated and downregulated genes relative to the normal cells. These gene targets of OT overlapped with the DEGs of various BC subtypes (e.g., LaBC, BBC, Her2BC, and TNBC) and identified 34, 47, 49, and 50 overlapping gene targets for upregulated and 51, 55, 67, and 68 for downregulated genes, respectively. Topological analyses in cytoHubba, specifically MCC, MNC, degree, and closeness, were applied to both upregulated and downregulated genes to determine the hub genes for each BC subtype. As presented in Figure 4, OTxLaBC has 8 upregulated DEGs (*PIK3R1*, *PIK3CA*, *EGFR*, *KIT*, *PDGFRA*, *FGFR1*, *MET*, *INSR*) and 8 downregulated DEGs (*STAT3*, *ESR1*, *MMP9*, *MAPK1*, *HSP90AB1*, *NFKB1*, *PIK3CA*, IGF1R); OTxLbBC has 8 upregulated DEGs (*EGFR*, *PIK3CA*, *PIK3R1*, *KIT*, *ALB*, *BCL2*, *APP*, *PDGFRA*) and 7 downregulated DEGs (*AKT1*, *ESR1*, *MMP9*, *STAT1*, *HSP90AB1*, *MAPK1*, *GSK3B*); OTxHer2BC has 8 upregulated DEGs (*EGFR*, *PIK3R1*, *PIK3CA*, *ESR1*, *IGF1R*, *BCL2*, *INSR*, *PDGFRA*) and 8 downregulated DEGs (*AKT1*, *STAT1*, *MMP9*, *MAPK1*, *GSK3B*, *HSP90AB1*, *PTPN11*, *CDK1*); and lastly, OTxTNBC has 8 upregulated DEGs (*PIK3CA*, *PIK3R1*, *ESR1*, *IGF1R*, *KIT*, *BCL2*, *MME*, *INSR*) and 8 downregulated DEGs (*STAT3*, *SRC*, *HSP90AB1*, *STAT1*, *GSK3B*, *MAPK1*). For further analysis, the hub genes of each subtype were overlapped to identify a common key hub gene that interacted with OT compounds, which were *MAPK1* and *HSP90AB1* for downregulated DEGs, while *PIK3R1* and *PIK3CA* for upregulated DEGs. In these collections of hub genes, variations in regulation were also observed across BC subtypes. For example, *ESR1* was downregulated in LaBC and LbBC but upregulated in Her2BC and TNBC, and *IGF1R* was downregulated in LaBC but upregulated in Her2BC and TNBC.

The overlapped genes were analyzed using DAVID to obtain functional annotations based on their BP, CC, and MF. Figure 5 and Appendix A present the annotated deregulated genes for each subtype, displayed with the log-transformed values of the highest fold enrichment (FE) from the functional annotations. All upregulated genes across the BC subtypes shared common annotations, with cellular response to jasmonic acid stimulus for BP, receptor complex for CC, and phenanthrene 9,10-monooxygenase activity of MF. In contrast, the downregulated genes across BC subtypes showed unique functional annotations in GO analysis. For BP, OT electron-shuttling compounds targeted biological processes for different BC subtypes, such as regulation of signaling receptor activity for LaBCxOT, cellular response to UV-A for LbBCxOT, collagen catabolic process for Her2BCxOT, and mitotic nuclear membrane disassembly for TNBCxOT. In terms of CC, the subcellular or extracellular locations that the OT compounds target the genes related to BC subtypes include dendritic spine for LaBCxOT, extracellular matrix for LbBCxOT, spindle pole for Her2BCxOT, and spindle microtubule for TNBCxOT. Lastly, GO analysis revealed key MFs for the OT compounds targeting the downregulated genes of different BC subtypes through insulin receptor substrate binding for LaBCxOT, nitric oxide synthase regulator activity for LbBCxOT, protein serine/threonine/tyrosine kinase activity for Her2xOT, and cyclin-dependent protein serine/threonine kinase activity for TNBCxOT.

### 3.5. Molecular Docking

Molecular simulations were performed to validate the network pharmacology findings, suggesting compounds in OT as chemopreventive agents through the interaction of the identified electron-shuttling ligands with exceptional pharmacokinetics and pharmacodynamics profiles with the hub genes from the BC subtypes. The proteins encoded by the identified hub genes were analyzed in the context of their functional interactions. From the initial collection of 265 compounds in OT, 22 bioactive compounds that have passed through various criteria were identified and docked to the crystal structures of such proteins, which are as follows: ESR1, HSP90AB1, IGF1R, MAPK1, PIK3CA, and PIK3R1. Gold-standard drugs with established inhibitory effects against the oncoproteins in BC tumorigenesis, which are sorafenib, alpelisib, tamoxifen, ganetespib, and linsitinib, were used as controls for comparison to the experimental ligands, with their binding affinities listed in Appendix A. Sorafenib is a multikinase inhibitor that targets angiogenesis and BC cell proliferation by inhibiting VEGFR, PDGFR, as well as RAF kinase [86], a primary mediator of the MAPK pathway [87]. Alpelisib is a selective PI3Kα inhibitor, specifically PIK3CA, that disrupts the PI3K/AKT/mTOR pathway responsible for cell growth, survival, and metabolism [88]. Tamoxifen is a well-established selective estrogen receptor modulator (SERM) that exhibits both antagonistic and agonistic estrogen signaling in different parts of the body, but mainly functions as an antagonist to present antitumor effects against hormone receptor-positive BC cells [89]. Ganetespib is a second-generation HSP90 inhibitor that destabilizes the oncogenic client proteins of HSP90 critical for signal transduction, promoting cancer cell survival and proliferation, including tyrosine kinases, HER2, and mutant p53 [90]. Linsitinib is a potent competitive IGF-1R/insulin receptor (IR) inhibitor in the autophosphorylation activities of the IGF ligands (e.g., IGF-I and IGF-II), impeding the activation of downstream cascades favoring BC tumorigenesis (e.g., AKT and ERK signaling) [91]. As presented in Figure 6 and detailed in Appendix A, all candidate ligands have a strong binding affinity towards the proteins encoded by the hub genes of BC subtypes. The lower binding affinity scores indicate strong protein–ligand interactions. Interestingly, several polyphenolic compounds demonstrated stronger interactions with MAPK1—specifically cyanidin, delphinidin, leucocyanidin, luteolin, quercetin, and tri-cetinidin—surpassing the binding performance of the control drug sorafenib. For ESR1, all experimental compounds showed superior interactions compared to the tamoxifen, with the exception of catechol, ascorbic acid, gallic acid, and 1-(3,4,5-trihydroxyphenyl) ethenone Among the tested ligands, phenolic acids (e.g., gallic acids, caffeic acid, catechol, 1-(3,4,5-trihydroxyphenyl)ethenone) had relatively higher binding affinity scores than the flavonoids, suggesting weaker protein–ligand interaction. Notably, flavonoids, such as catechins, delphinidin, fisetin, quercetin, tricetinidin, cyanidin, and (-)-epigallocatechin, exhibited favorable binding profiles. Luteolin was remarkable due to its strong binding to both *MAPK1* for the downregulated genes and *PIK3CA* for upregulated genes across all BC subtypes, with binding scores of −9.1 and −8.4 kcal/mol, respectively. The 3D superimposed graphical representations of the control drugs and the top three experimental ligands with exceptional binding affinity to hub proteins are shown in Appendix A.

Given the promising thermodynamic performance of luteolin, further molecular investigations were performed using pharmacophore mapping to elucidate its interaction with the key hub proteins associated with BC subtypes. As illustrated in Figure 7, the substituents in luteolin were found to interact with the specific amino acid residues within the receptors of the protein, significantly contributing to its strong binding affinity to these hub proteins, which are as follows: for MAPK1 residues, Ala46, Ile25, Leu150, Cys160, Asp161, Gln99, Val33, Lys 48; and for PIK3CA residues, Met816, Ile826, Ile694, Val744, Ile742, Asp827, Val745, Tyr730, and Asp704. The stabilization of the protein–ligand complexes was primarily attributed to the extensive hydrogen bonding by the hydroxyl substituents of luteolin. These interactions were complemented by the synergistic network of various non-covalent interactions, including π–sigma, π–alkyl, π–sulfur, and π–π interactions, which contribute to the strengthened binding interaction. Such interactions underscored the molecular mechanism of protein–ligand binding by the chemical and structural properties of luteolin and the binding sites, respectively.

### 3.6. Molecular Dynamics

Since the principle of structural biology posits that the structure of the protein fundamentally determines its function, molecular dynamics simulations were conducted to investigate the conformational changes of the key hub proteins upon binding with electron-shuttling compounds. Here, a 200 ns molecular dynamics simulation was conducted for both ligand-unbounded (apo state) and ligand-bounded (holo protein) forms of the proteins using luteolin, which exhibited the highest binding affinity for MAPK1 and PIK3CA. The analyses of these two states included RMSD, RMSF, Rg, SASA, and the number of hydrogen bonding profiles, as shown in Figure 8.

The results revealed significant conformational fluctuations in the holo protein compared to its apo counterparts. RMSD is a measure of structural deviations in the protein backbone between the initial and final structural conformations [92], serving as an indicator of protein stability. Smaller deviations in the protein state indicate a more stable protein conformation. In the MAPK1 systems, the holo protein exhibited a slightly higher RMSD profile compared to the apo protein during the latter part of the simulation. The apo protein initially fluctuated between 1.87 and 4.32 Å from 42.2 to 14.18 ns, then stabilized around 2.94 Å from ca. 116 to 200 ns. In contrast, the holo protein exhibited a deviation up to 3.33 Å at 20.55 ns, followed by a drop to 1.65 Å at 59.53 ns. Afterward, it gradually increased to 4.43 Å by 143.96 ns and stabilized at around 3.28 Å toward the end of the simulation. Similarly, for PIK3CA, notable discrepancies were observed between its apo and holo states during the 200 ns simulation, with a crossover point at approximately 169.59 ns where the RMSD of the holo state exceeded that of the apo. Before this point, the apo protein exhibited more aggressive fluctuations and higher RMSD values, while the holo protein showed a more stable profile. From 20 ns to 150 ns, the RMSD of the holo protein remained relatively stable, increasing slightly from ca. 2.96 to 3.14 Å. It then rose to ca. 4.61 Å at 175.33 ns and stabilized at 4.06 Å until the end of the simulation. These results indicate that the holo protein underwent a continuous conformational shift, ultimately reaching a higher deviation compared to the apo form by the end of the simulation.

Rg measures the compactness of a protein by reflecting the overall spatial distribution of its atoms [93]. A higher Rg value indicates a less compact structure. In MAPK1, similar Rg profiles were observed between the apo and holo states, although some differences emerged during specific time frames. The holo MAPK1 protein exhibited a relatively stable and higher Rg (approximately 22.54 Å from 163.82 to 200 ns) compared to the apo state, which stabilized around 22.20 Å from 100 to 200 ns. During the initial 100 ns, the apo protein displayed erratic fluctuations before stabilizing. Meanwhile, the holo protein showed a gradual increase in Rg, peaking at 23.08 Å at 143.75 ns before briefly dropping to 22.19 Å between 151.20 and 162 ns. In contrast, PIK3CA demonstrated more distinct dynamic differences between its apo and holo forms. The apo protein showed distinct fluctuations in Rg across different time windows: (1) a peak at 31.86 Å at 29.98 ns; (2) a stable decrease to 31.48 Å between 34.30 and 65.28 ns; (3) an increase to 32.20 Å at 80.25 ns; (4) a sharp drop to 31.31 Å at 100.24 ns, followed by (5) a rapid rise to 32.30 Å between 100 and 138.36 ns, and finally, (6) a gradual decrease to around 31.70 Å from 140 to 200 ns. On the other hand, the holo PIK3CA protein exhibited a gradual and continuous decrease in Rg, from 31.52 to 31.32 Å, through the simulation, indicating increasing structural compactness over time. Overall, these variations suggest distinct dynamic behaviors between the apo and holo states of both proteins. In MAPK1, the apo state experienced more fluctuations early on but stabilized later, while the holo state maintained a more consistent Rg after equilibration. For PIK3CA, the apo state exhibited dynamic shifts across multiple time frames, whereas the holo protein underwent a steady decrease in Rg, implying a continuous structural compaction throughout the simulation.

SASA represents the region of a protein exposed to solvent molecules as a result of van der Waals contact surface of the molecules, providing insights into protein folding and stability [94,95]. For MAPK1, the apo and holo states showed a stable SASA trend throughout the simulation, maintaining values around 19,000 Å^2^/N. In the larger PIK3CA protein, the SASA of the holo state remained relatively stable with fluctuations around 45,800 Å^2^/N compared to the apo state (~46,000 Å^2^/N). The exposed surface area to the solvent initially increased from approximately 44,200 Å^2^/N at 0 ns to around 47,800 Å^2^/N at 13 ns. It then began to equilibrate at ~45,700 Å^2^/N between 30 ns and 120 ns. This was followed by an increase to 48,100 Å^2^/N at 140 ns, after which the SASA fluctuated until 170 ns, eventually decreasing to ~46,000 Å^2^/N by the end of the simulation. These differences highlight the effect of ligand binding on solvent accessibility, potentially influencing protein stability and functional interactions.

Hydrogen bonding is a crucial intermolecular force governing electron donation and acceptance within the protein, significantly contributing to protein stability and folding [96]. For MAPK1, the interaction with luteolin induced the formation of a total of 0–7 hydrogen bonds in the protein throughout the simulation, occurring with frequencies of 510, 4307, 7807, 5029, 1765, 529, 53, and 1 for 0, 1, 2, 3, 4, 5, 6, and 7 hydrogen bonds formed, respectively. On the other hand, the luteolin-bound PIK3CA formed a robust hydrogen bonding 0, 1, 2, 3, 4, 5, and 6 hydrogen bonds between luteolin and the binding pocket of PIK3CA, occurring with frequencies of 1768, 7515, 6745, 3027, 826, 115, and 5, respectively. This dynamic hydrogen bonding pattern suggests that ligand binding can either stabilize or disrupt key interactions, depending on the structural and functional context of the protein.

RMSF analysis was conducted to assess the flexibility of the key hub proteins across BC subtypes in the presence of a ligand. RMSF quantifies the fluctuation of each residue throughout the molecular dynamic simulation, corresponding to the stiffness and flexibility of specific regions by comparing apo and holo proteins of key BC hub proteins. A high RMSF value indicates greater flexibility in a particular amino acid residue or region, whereas a low RMSF suggests rigidity. Notably, the formation of the protein–ligand complexes stabilizes the protein itself, leading to lower total energy and RMSF values in the holo form compared to its apo counterpart. Consistent with this notion, the RMSF profiles revealed that, for most residues, the holo form exhibited reduced fluctuations relative to the apo form. When considering significant percent changes in RMSF values (% change > ±30%), ligand binding resulted in both increased and decreased fluctuations across residues (see Appendix A), indicating shifts in protein flexibility and rigidity. Specifically, the luteolin binding to MAPK1 led to an increased fluctuation in the amino acid residues of ca. 7.34% (26 out of 354), and decreased fluctuations for 9.04% (32 out of 350 residues). In contrast, the interaction of luteolin with PIK3CA induced greater fluctuations in 10.47% (99 out of 946 residues), alongside reduced fluctuations in 7.93% (75 out of 946 residues) in other regions of the protein. Looking closely at the binding site residues (see Appendix A), a decrease in the change of fluctuations in the MAPK1 residues and a combination of increased and decreased residue fluctuations in the PIK3CA residues were observed in the local RMSF profiles. This variation in flexibility may influence the overall binding stability and functional implications of the ligand–receptor interaction. These findings entail the intricate effects of ligand binding on protein dynamics, supporting the role of luteolin in modulating structural flexibility at a molecular level.

The ΔG for each complex was computed using the MM/PBSA method to validate the molecular dynamics simulations, focusing on stable trajectory-based data within the time windows 50–70 ns for MAPK1 and 100–120 ns for PIK3CA, as shown in Figure 9. For the MAPK1 complex, the total enthalpy change was calculated to be −22.94 kcal/mol, with a corresponding entropy contribution of 7.42 kcal/mol, resulting in a final ΔG of −15.52 kcal/mol. For the PIK3CA complex, the total enthalpy change was −17.79 kcal/mol, with an entropy term of 9.44 kcal/mol, yielding a ΔG of −8.35 kcal/mol. The decomposition of the enthalpic terms—including van der Waals, electrostatic, polar solvation, and nonpolar solvation energies—is provided in Appendix A.

The PCA/FEL analysis of the MAPK1–luteolin complex shows dynamic shifts in conformational stability across time windows. From 0–20 ns (Figure 10A), the free–energy landscape is dominated by a single predominant energy minimum, with PC1 and PC2 ranging approximately from −5 to 5 nm. At 50–70 ns (Figure 10B), the conformational space becomes more dispersed, extending up to ±15 nm along PC2 and between −10 nm and −8 nm along PC1, indicating increased structural variability. By 180–200 ns (Figure 10C), the system returns to a more compact energy landscape with a single predominant minimum, with PC1 spanning −5 to 5 nm and PC2 ranging from −5 to 3 nm. For the PIK3CA–luteolin complex, the initial time window of 0–20 ns (Figure 10D) reveals a single, well-defined global energy minimum, localized within PC1 and PC2 values of −6 to 0 nm, surrounded by scattered regions of moderate free energy. At 100–120 ns (Figure 10E), the energy landscape becomes more heterogeneous, showing three distinct local minima. By 180–200 ns (Figure 10F), the global minimum shifts position, now centered around PC1 values of −2 to 4 nm and PC2 values of −4 to 3 nm, again surrounded by scattered moderate free–energy regions.

## 4. Discussion

### 4.1. Mechanism of Oolong Tea Against Breast Cancer

BC encompasses a variety of malignant tumors that develop in the mammary glands [97], with the characteristics of the hallmarks of cancer, such as maintaining proliferative capacity, inducing angiogenesis, evading growth suppressors, resisting cell death, achieving replicative immortality, invading adjacent tissues, metastasizing to distant sites, evading immune attack, exhibiting genomic instability, promoting inflammation, and sustaining energy production [98]. As mentioned previously, BC cases are typically caused by BRCA1 and BRCA2 gene mutations and risk factor exposure, such as hormonal aspects, physiological conditions, nutritional influences, and lifestyle choices [99]. Paradoxically speaking, while DNA damage induces cancer cell death with the use of exogenous chemicals or radiation, it could also promote BC tumorigenesis through exacerbation of the gene mutation frequency [100]. Seeing that a good balance between free radicals and antioxidants may play a significant role in immune responses, the accumulation of the free radicals increases the oxidative stress in cells, which may be the culprit for not only inducing DNA damage [101], but also promoting inflammation, inactivating tumor suppressors, promoting genetic instability, and stimulating signaling pathways related to BC progression [102]. Such an imbalance of free radicals to antioxidants highlights the use of plant-derived bioactive compounds with antioxidant and electron-shuttling properties, in lieu of current expensive and invasive treatments (e.g., mastectomy, radiation, chemotherapy, and hormone therapy [103]), to aid in the attenuation of BC tumorigenesis and progression.

For a long time, the consumption of tea beverages provided a great pharmaceutical benefit, including anti-inflammatory [104], antiviral [105], chemopreventive and, potentially, anti-cancer activity, due to its high polyphenol content. Experimental data have shown that the natural products in different fermentation times of *Camellia* tea extracts, specifically OT, possess great antioxidant and anticancer activities and inhibitory potential on the growth of in vitro cancer cell lines in a dose-dependent manner [106]. Shi et al. assessed different degrees of tea fermentation against different BC subtypes and presented that green tea and OT inhibit cell proliferation across all BC subtypes by promoting DNA damage and cleavage and reducing the proliferative capability of BC cell colonies on soft agar [32]. This may be linked to their electron-shuttling properties, which can be quantitatively measured by MFC and assessed by CV, in maximizing bioenergy stimulation for enhanced electron transfer and ATP production. For a shorter time of tea fermentation, the low dosage consumption of white tea in BC-induced rat models demonstrated anti-inflammatory effects, lipid peroxidation and DNA damage prevention, apoptosis-induced tumor suppression, and NRP-2 activity and M-CSF release inhibition [107]. As the MOA of OT against BC subtypes is being unraveled, this study, via in vitro and in silico approaches, suggests that electron-shuttling compounds in OT can act as an electrochemical catalyst by neutralizing free radicals and maximizing mitochondrial functions through bioelectricity generation and as a chemopreventive agent by interacting with key hub proteins of respective BC subtypes.

### 4.2. Phytochemical Content Assays

The phytochemical content in several plant extracts and traditional Chinese medicines has been identified as a key player in eliciting bioactivity [104,105,108], including the chemopreventive potential of *Camellia* tea [109]. To assess the efficacy of OT as an antioxidant and anticancer, TPC, TFC, and TCTC in OT extracts were quantitatively evaluated. In this study, OT underwent two different extraction methods—solvent-extracted OT (OTL) and supercritical fluid-extracted OT (OTS)—to extract the phytochemical contents from OT. These differences in extraction method and solvent were applied to scrutinize which extract would present bioactivity. As shown in Table 2 different extraction methods of OT using two solvent systems yielded significant amounts of phytochemical contents, which constitute their physiological effects such as antioxidant, electron-shuttling, and chemopreventive properties. SE and SFE were employed for OT extraction, using water/ethanol (polar solvents) and supercritical CO_2_ (a non-polar solvent), respectively. Among the OT extracts, OTL-E with 95% aqueous ethanol exhibited the highest phytochemical content (i.e., TPC, TFC, and TCTC) compared to the water-based extracts, OTL-W and OTS-W. This indicates that the efficiency of polyphenol extraction depends on the polarity of the solvent used. SE with ethanol exhibited the optimal conditions for extracting polyphenols from OT due to its dual polarity, effectively solubilizing a wide range of bioactive compounds. In terms of extraction technique, OTS extracts yielded lower polyphenol content due to the limited ability of low-polar supercritical CO_2_ to extract non-polar and moderately polar compounds [110]. While polar solvents effectively extract hydrophilic compounds (e.g., polyphenols and flavonoids), nonpolar solvents are more suited for hydrophobic compounds (e.g., essential oils and phenols). The combination of water and other organic solvents (i.e., 95% aqueous ethanol) has been shown to improve the extraction efficiency of polyphenols compared to using solely pure solvents. In fact, OTL-W extract had a substantial number of phytochemicals with TPC and TFC values of 31.93 ± 0.49 g GAE/100 g and 16.44 ± 0.53 g QE/100 g, respectively [111]. However, similar results were obtained from the paper of Šeremet et al., where a comparative analysis of TPC levels in OT extracted using water and 50% aqueous ethanol at varying pressure and holding times revealed that OTL-E had a higher TPC content than OTL-W, despite pressure variations [112]. Do et al. investigated the effects of different extraction solvents on optimizing the TPC, TFC, and antioxidant activity in the *Limnophila aromatica*, indicating that ethanol was a suitable solvent for extracting polyphenols with TPC values of 40.50 ± 0.88, 30.60 ± 1.36, and 30.30 ± 0.54 mg GAE/g; and TFC of 31.11 ± 0.43, 19.47 ± 0.35, and 17.19 ± 0.15 mg QE/g for 100% ethanol, 75%, and 50% aqueous ethanol, respectively [113]. In contrast, water as a solvent yielded significantly lower TPC and TFC values with 6.25 ± 0.24 mg GAE/g and 4.04 ± 0.08 mg QE/g, respectively. Such findings align with the study of Nguyen et al. [114], which demonstrated that 90% aqueous ethanol resulted in higher TPC and TFC yields in *Polyscias fruticosa* roots, with values of 28.56 µg GAE/mg and 26.26 µg QE/mg, respectively, compared to those of 70% and 50% ethanol.

### 4.3. Antioxidant Activities

An imbalance favoring free radicals over antioxidants disrupts homeostasis, favoring a cascade of physiological dysregulation. This oxidative stress can lead to DNA damage, suppressing key enzymes and activating signaling pathways that promote BC development. Introducing phenolic compounds as exogenous antioxidants may help mitigate such oxidative stress and restore the balance. As shown in Table 3, OTL-E demonstrated the highest antioxidant potential in DPPH and FRAP assays compared to the water extracts, OTL-W and OTS-W. This result is consistent with the substantial number of phytonutrients in OTL-E (TPC and TCTC), which significantly enhance the potency of bioactive compounds as antioxidants. Flavonoids and tannins, subclasses of polyphenols, serve as exogenous antioxidants and contribute to the medicinal benefits of plants, including antioxidant, anticancer, and anti-inflammatory properties [115,116,117,118,119,120]. The antioxidant property lies in their chemical structure, which is the presence of highly reactive phenolic hydroxyl (-OH) groups that donate hydrogen atoms or electrons to neutralize free radicals, forming more stable and less reactive species [121]. This free radical neutralizing process involves the inhibition of nitric oxide synthase (NOS) and xanthine oxidase activities, modulation of channel pathways, and is governed by HAT, SET, and sequential proton-loss electron transfer [122,123]. Lin et al. investigated the antioxidant and anti-inflammatory properties of polyphenols in OT extracts [124]. They reported that the OTL-W had TPC of 104.6 mg GAE/g dry weight (DW), TFC of 11.9 ± 0.4 mg CE/g DW, and TCTC of 8.2 ± 1.3 cyanidin chloride equivalent/g DW, which were sufficient to exhibit antioxidant potency with an IC_50_ of 57.2 ± 6.0 μg/mL in the DPPH assay and anti-inflammatory effects by inhibiting the NO production by 60%, though it did not directly suppress the iNOS and COX-2 expression in RAW264.7 cells.

Catechins are naturally occurring polyphenols, specifically flavonols, a subfamily of flavonoids, in *Camellia* tea, possessing two benzene rings and a dihydropyran ring with hydroxyl groups at C3 [125]. Such a compound possesses a strong antioxidant activity via scavenging free radicals through the donation of an electron from the phenolic-rich hydroxyl groups to stabilize the reactive oxygen and nitrogen species [126]. The degree of the antioxidant potential of hydroxyl-rich compounds is subject to the number and arrangement of the hydroxyl groups and the extent of structure conjugation [125]. Catechin undergoes esterification with its gallate groups, forming catechin derivatives (e.g., epicatechin gallate, epigallocatechin, and epicatechin gallate). These compounds, especially epicatechins, polymerize to form a type of condensed tannin—proanthocyanidins or procyanidins, through linking 4→8 carbon–carbon (for A- and B-type procyanidins) and a 2→O7 ether (for A-type procyanidins) bonds [127]. As shown in Table 2, among all extracts, OTL-E had the highest amount of condensed tannin, which corresponds to the polymerized catechins and derivatives, suggesting its role in high antioxidant activity in both DPPH and FRAP assays (see Table 3). This result was consistent with the paper by Šeremet et al. [112], revealing aqueous 50% OTL-E (ca. 9–17 mmol Trolox/mL) had greater antioxidant activity compared to that of water-extracted OTs (ca. 6–13 mmol Trolox/mL) despite the variations in extraction times and pressure level. In fact, the antioxidant activity contributes to the chemopreventive properties of plants through decreasing oxidative stress and cellular damage [128]. Mittal et al. found that the dose-dependent treatment of EGCG has been shown to inhibit the cell proliferation and viability of human breast carcinoma MCF-7 cell lines due to apoptosis as a result of the inhibition of telomerase activity and human telomerase reverse transcriptase [129]. The same compound enhanced the 5-fluorouracil-induced cytotoxicity in colorectal cell lines, promoted apoptosis and cell cycle arrest to inhibit the drug-resistant cell lines, and suppressed self-renewal proteins Notch1, Bmi1, Suz12, and Ezh2 expressions, stimulating the tumor suppressive micro RNAs [130]. Other catechin derivatives (e.g., methyl gallate and catechin-3-O-gallate) from *Acacia hydaspica* as chemopreventive agents exhibited a dose-dependent inhibition of TNBC MDA-MB-231 cell lines in terms of the cell survival and proliferation through modulating several cell signaling oncoproteins (e.g., BCL2, MAPK, PI3-K, JAK2, and STAT3) [131]. Notably, proanthocyanidins work synergistically with histone deacetylase inhibitor drugs to promote apoptosis and, at the same time, to inhibit the growth and proliferation of MDA-MB-231 cell lines [132].

### 4.4. Electrochemical Analysis

As prior serial studies revealed, electrochemical characteristics (e.g., electrochemical catalysis of ESs) were important in persisting and/or stimulating disease-treating efficacy, likely due to sustained MOA via the electron transport chain for redox-mediating medication. Thus, to exhibit overall expression in bioenergy stimulation, MFCs have been widely utilized to investigate the electrochemical properties of electron-shuttling compounds concerning their therapeutic potential derived from various Chinese herbal medicines and plant foods [84,104,105]. Among all extracts, OTL-W at 2000 ppm exhibited the highest PD amplification with 2.47 ± 0.54-fold compared to the blank (see Figure 1 and Table 4). Previous studies have suggested that the amplification factor above 2.00 indicates the presence of ESs due to significant electron transfer capabilities, which are closely linked with physiological effects, such as chemopreventive potential [13,33]. The unique structure of ESs (i.e., *ortho*- or *para*-dihydroxyl substituents on aromatic structures) contributes to their role in redox activities [14,15]. The resonance effect facilitated by lone pairs in oxygen atoms enhances electron acceptance or donation through alternative reaction routes, lowering the activation energy and increasing reaction rates [15]. Unlike amino substituent-owning ESs that are toxic to certain organisms [133], compounds with hydroxyl substituents acting as electrochemical catalysts, such as flavonoids and condensed tannins, from tea extracts were found to provide stability, reversibility, and biocompatibility, rendering them sustainable for pharmaceutical applications [15,134]. Due to their reversible reduction and oxidation ability, ESs serve as redox mediators, facilitating electron transfer across multiple redox reactions.

The extraction method significantly influences the yield of phytochemical concentration in OT extracts, which is correlated with its electrochemical properties, affecting their PD profiles in MFC systems. OTL samples demonstrated better MFC performance compared to OTS extracts, as the latter primarily isolates less polar polyphenols (e.g., vanillic acid, syringic acid, and ferulic acid derivatives) using non-polar, supercritical CO_2_ [135]. Consequently, the reduced polyphenol yield from SFE led to lower PD values in MFC, emphasizing the role of hydroxyl-rich polyphenols for optimal electron-shuttling activity. Furthermore, ethanol extracts have been identified as promising candidates for bioactivity due to their ability to extract a broad range of polyphenols [13]. However, OTL-E exhibited toxicity to electroactive, Gram-negative bacteria *A. hydrophila* at higher concentrations, with the highest PD at 250 ppm before declining at increased concentrations. This may be due to SE extracting certain phytochemicals to a level that becomes toxic to electroactive bacteria in MFC systems. This toxicity effect parallels ethanolic green tea extracts, which showed antibacterial and bactericidal activity against Gram-positive bacteria (*Bacillus subtilis*, inhibition zone: 4.3 ± 0.3 mm) and Gram-negative bacteria (*Pseudomonas aeruginosa*, inhibition zone: 1.5 ± 0.3 mm), with MIC/MBC values of 25/50 µg/mL for *B. subtilis* and 6.25/12.5 µg/mL for *P. aeruginosa*, respectively [136]. Given *B. subtilis* and *A. hydrophila* share common electroactive properties, ethanolic green tea demonstrated immunostimulatory effects in *Labeo rohita*, increasing post-challenge survival (76%) against *A. hydrophila* and reducing infection progression [137]. Similarly, Kuo et al. reported that green tea residues exhibited antibacterial activity against *A. hydrophila*, with MIC and MBC values of 0.78 and 1.56 mg/mL, respectively [138]. In fact, some natural polyphenols and flavonoids in *C. sinensis* showed antibacterial properties against *A. hydrophila*, influencing their bioenergy-stimulating properties through MFC [15]. Despite catechin displaying remarkable bioactivity, at some concentrations, it exerts an antibacterial effect on tea extracts through perforation and fluidity reduction of the bacterial cell membrane, thereby destroying the cell that leads to internal electrolyte leakage [139]. As studies on the antimicrobial mechanisms of phytochemicals remain limited, further investigation is needed to elucidate how the concentration of each phytochemical compound in ethanol extracts influences their performance in MFC systems.

To explore detailed electrochemical characteristics of herbal samples, CV was employed alongside MFC to identify electron-shuttling compounds in tea extracts by analyzing their reversible electron transfer facilitating redox activities [104]. ESs exhibited sustained bioelectrochemical activity with both reduction and oxidation potentials, reflected as distinct peaks in their respective voltammogram regions, which differentiated them from antioxidants, characterized by shorter durations of dominance and irreversible oxidative potentials appearing exclusively in the oxidation region [4]. Moreover, as serial CV also simulated repeated oxidation and reduction to take place on test samples, nearly identical responding closed-loop profiles could elucidate their electrochemically stable and reversible nature as electrochemical catalysts. The CV profiles of OT water extracts confirmed the presence of ESs with both reduction and oxidation potential peaks, indicating reversible electron transfer between oxidized and reduced states (e.g., catechins and their derivatives), whereas antioxidant activity was more pronounced in OT ethanolic extracts, where certain compounds (e.g., gallic acid) acted solely as reducing agents with irreversible oxidation activities [140]. Their attenuation of AUCs was linked to the irreversible redox activities of electron donors and acceptors, with the remaining potential peaks after attenuation likely attributed to ESs that facilitated reversible redox reactions; these compounds initially underwent reduction and then stabilized asymptotically, exhibiting electrochemical catalytic activity and enhanced electron transfer [141]. The performance of OT extracts in CV aligned with their MFC profiles, as higher CV areas at the original pH levels of the extracts corresponded to increased bioelectricity generation in MFCs at 750 ppm, with the ranking as follows: OTL-W (CV: 99.986 μW, MFC: 2.04 ± 0.25) > OTS-E (CV: 45.215 μW, MFC: 1.80 ± 0.29) > OTL-E (CV: 37.156 μW, MFC: 1.56 ± 0.33) > OTS-W (CV: 14.359 μW, MFC: 1.40 ± 0.27). These enhanced electrochemical profiles of OT extracts in CV were due to the use of GCE with carbon black, which is known for its excellent conductivity and high surface area. The modified electrode provides a more efficient surface for the adsorption of electron-shuttling compounds, allowing for better interaction between the electrode and the electroactive species in the tea extracts and, thus, improving electron transfer properties [142,143]. This modification provided a better resolution to allow better differentiation between the redox activities of antioxidants and electron-shuttling compounds. In fact, several studies reported redox potential peaks ranging only from ca. 5.00 to −1.00 μA for OTL-W extracts at their native pH using unmodified GCE [140,144]. In comparison, the current study observed significantly higher redox peaks for the same extract using modified GCE, reaching ca. 150 to −75 μA. Despite high phytochemical contents across all extracts, variations in redox activity were attributed to differences in extraction methods, solvents, and pH conditions, which influenced the yield and composition of bioactive compounds, ultimately affecting their ability to facilitate redox activities effectively.

As mentioned elsewhere, the electrochemical performance of medicinal herbs and tea extracts in CV is pH-dependent [141]. The CV profiles of the OT extracts in their native pH (pH 4.20–4.53) displayed the largest area, indicating the most favorable conditions for redox reactions. This implies that the OT bioactive compounds with electron-shuttling and antioxidant capabilities exhibited the highest electrochemical activity at their original pH due to their optimal ionization (i.e., protonation and deprotonation) and their ability to undergo reversible redox reactions, allowing efficient electron transfer in both directions [145]. However, as pH levels increase, the hydroxyl groups in polyphenols undergo ionization through deprotonation and protonation [146]. In this process, the hydroxyl group donates a hydrogen ion to hydroxide ions, forming water and converting the polyphenol into its conjugate base. This reaction leads to the electrooxidation decomposition of electrochemically active compounds (i.e., antioxidants producing irreversible, unstable intermediates) while maintaining the reversible and stable redox curve potentials of electron shuttles. The deprotonation of antioxidants is linked to a reduction in the closed-loop CV area, emphasizing the crucial role of electron shuttles in maintaining redox activity due to their stability regardless of pH level [15,146]. There was a noticeable attenuation in the CV profiles when the pH was increased to ca. 8.5. Most OT extracts, except water extracts, retain electrochemical activity but favor irreversible oxidation, shifting away from reversible electron transfer behavior. Despite this, some compounds retain electrochemical activity, albeit with diminished reversibility, likely due to changes in solubility and reactivity that reduce mass transfer resistance and enhance electron transport [15]. At a highly alkaline pH of ca. 10, most of the CV profiles of OT extracts presented diminished electrochemical activity and primarily oxidative potentials. Interestingly, the OTL-W extract was maintained to exhibit a pair of reduction and oxidation peaks even at pH 10, suggesting that the compounds present in this particular extract might be more stable to present as an electrochemical catalyst and less affected by high pH conditions. This behavior could be related to the specific composition of the water extract, where certain polyphenols may remain active across a wider pH range, facilitating both reduction and oxidation processes [15].

Given the electron-shuttling properties of compounds in OT as redox mediators, which enhance electron transfer efficiency for bioelectrogenesis, these compounds may play a crucial role in intracellular bioenergy metabolism, specifically the flow of electrons from Kreb’s cycle to the electron transport chain in the mitochondria [147]. Polyphenols may enhance oxidative phosphorylation for ATP production by accelerating electron transfer between the intercellular compartment and extracellular medium, thereby satisfying cellular energy demands [148]. However, in cancer cells, this metabolic activity is disrupted due to mitochondrial dysfunction along the electron transport chain, leading to the following consequences: (1) increased glutamine and glucose uptake; (2) reduced current in the electron transport chain, (3) increased concentrations of reduced cofactors; and (4) the Warburg effect, characterized by a redox shift from oxidative phosphorylation to glycolysis for ATP generation despite the presence of oxygen [149]. This metabolic reprogramming exacerbates tumorigenesis by increasing lactate production as a byproduct, leading to acidification of the tumor microenvironment [150]. Notably, OT extracts have demonstrated the ability to maximize bioelectrogenesis by stimulating mitochondrial activity, as observed in *A. hydrophila* via MFC, potentially rooted in the chemical structure of electron-shuttling compounds (e.g., *ortho*- and *para*-dihydroxyl substituents) that act as catalysts in electron transfer. Among extraction techniques, OTL-W may be the most effective method for obtaining a sustainable, polyphenol-enriched source with anticancer potential to mitigate BC progression and enhance treatment outcomes.

### 4.5. Network Pharmacology

Herbal medicines are composed of a complex mixture of bioactive compounds that collectively elicit pharmacological effects [5,6,7,8,9]. The bioactive compounds in OT, known for their pleiotropic nature, target multiple genes and proteins involved in regulating key signaling pathways, impacting fundamental cellular processes such as proliferation, apoptosis, and differentiation in multifactorial diseases like BC. This inherent complexity provides OT with a broad range of therapeutic potential against various diseases but presents challenges in elucidating the specific MOA for a particular disease. To address this, network pharmacology was performed to holistically determine the MOA of the electron-shuttling compounds in OT with great pharmacokinetic and pharmacodynamic profiles as chemopreventive and possibly anti-cancer agents across different BC subtypes.

The topological analyses performed in cytoHubba revealed that the phytochemical compounds in OT play a significant role in targeting key hub genes across different BC subtypes through several mechanisms. One of the key MOAs of these compounds was modulating the critical signaling pathways (e.g., PI3K/AKT/mTOR, EGFR, and FGFR1) and receptor tyrosine kinases (e.g., c-Src, PI3Ks, EGFR, PDGF, KIT, INSR), all of which are crucial in BC tumorigenesis, proliferation, and metastasis. The hydroxyl-enriched flavonoids act as effective kinase inhibitors by interacting with these signaling proteins [151,152,153,154,155]. For instance, the luteolin binding to c-Src suppresses its phosphorylation and downstream activation of the Hippo signaling pathway, thereby reducing tumor progression and metastasis [156,157]. The upregulated PIK3CA and PIK3R1 across all BC subtypes, as key regulators of the PI3K/AKT/mTOR pathway, could potentially be modulated by these polyphenols, inhibiting this pathway through disruption of phosphorylation and conformational changes in the catalytic and regulatory subunits [157], alongside the regulation of the downstream effectors, such as STAT3 and STAT1. The downregulated CDK1 in Her2BC suggests a dysregulated cell cycle, where the bioactive compounds in OT could modulate through the disruption of the G2/M transition, leading to cell cycle arrest and thereby suppressing tumor cell proliferation [158]. Luteolin suppressed the HGF-MET-Akt pathway by interacting with the unphosphorylated MET to inhibit its activation, thus downregulating its protein expression and blocking the stimulus of the downstream Akt pathway [159]. Furthermore, EGCG further modulates EGFR-mediated hyperphosphorylation and activation of Ras/MAPK and PI3K/AKT/mTOR pathways by competitively binding with endogenous EGF ligands and ATP at the tyrosine kinase domain [160], thereby effectively reducing proliferation, DNA synthesis, and metastatic potential. The same catechin derivative exhibited a dose-dependent inhibition of PDGF-induced proliferation by blocking the ATP sites and reducing its autophosphorylation activity in the hepatic stellate cell lines [161], alongside its anti-angiogenic effects on cervical adenocarcinomas by downregulating PDGF gene expression [162]. These findings suggest that phytonutrients may intervene in signal transduction to support BC tumorigenesis, proliferation, and metastasis by modulating dysregulated genes and restoring them to homeostatic conditions.

The microarray data of BC subtypes also revealed dysregulation in genes responsible for apoptosis and cell survival, such as APP and BCL2. The upregulation of these genes corresponds to the cell survival, proliferation, migration, and apoptotic-resistant nature of BC cells by activating the AKT signaling pathway [163,164,165], where phytochemicals in OT could target these genes to modulate their overexpression, which reduces the downstream effects on BC progression. For instance, the luteolin-treated BC cell lines MDA-MB-231 and MCF-7 downregulated the APP gene expression by −4.25- and −3.41-fold change relative to control, respectively [166]. The dose-dependent treatment of delphinidin in human colon cancer downregulated BCL2, inducing apoptosis and thus reducing cell viability. The interactions between flavonoids and apoptotic signaling proteins may restore the balance in pro-apoptotic and anti-apoptotic signaling [167]. On the other hand, the downregulated AKT and GSK3B in the microarray data suggest that these genes were responsible for the increased BC cell survival due to impaired apoptotic activity. GSK3B is a key player in the Wnt/β-catenin, Hedgehog, and PIK3/AKT pathways that regulate cell proliferation, differentiation, migration, and apoptosis [168,169,170]; however, the dysregulation in its expression triggers the tumorigenic properties [170], leading to the apoptosis attenuation. The dysregulation in the gene expression across BC subtypes can be targeted by the OT phytonutrients to modulate their activity. Quercitrin in a dose-dependent manner impedes GSK3B activity in *Xenopus laevis* embryos, increasing the GSK3β S9 levels, and amplifying the Wnt/β-catenin pathway [171].

Furthermore, hormone receptor signaling, particularly mediated by ESR1, is another mechanism of OT compounds towards the development of BC subtypes. Results have shown that *ESR1* showed a dual dysregulation in BC. Such a gene was downregulated in luminal BCs due to the endocrine resistance through twist and snail as epithelial-to-mesenchymal transition (EMT)-related transcription factors, *ESR1* gene mutations, oncogenic signaling pathways (e.g., MAPK, CDK, PI3K/AKT pathways), and insulin receptor substrate-one (IRS-1) stimulation through IGF/insulin signaling pathway [172,173]. The *ESR1* expression in non-luminal BCs (i.e., Her2BC and TNBC) appeared upregulated despite the ER/PR absence. In Her2BC, the bidirectional crosstalk between HER2 and ER in BC could promote cell proliferation and survival through the phosphorylation of ERα and its co-regulators despite the absence of ER, which stimulates ER signaling pathways, such as downstream RAS, PI3K/AKT, and MAPK [174]. For TNBC with the absence of hormone receptors, the dual inhibition of Wnt and histone deacetylase (HDAC) may be the reason for the *ESR1* upregulation [175]. Furthermore, HSP90AB1 at normal conditions functions as a molecular chaperone to stabilize and activate a broad range of proteins through protein folding, including ESR1 [176], yet this was found upregulated across all BC subtypes that contributed to ER stress [177] while targeted by OT phytonutrients. Exogenous ligands from tea beverages with estrogenic and antiestrogenic properties could modulate this genetic dysregulation, promoting BC. Luteolin mimics methyl-p-hydroxyphenyllactate as a cell growth agent [178], thus acting as an antiestrogenic ligand through the regulation of several estrogen-signaling pathways [179,180]. In contrast, quercetin showed an estrogen-like activity on ER+ cell lines with greater activation of ERα (1.7-fold) and ERβ (4.5-fold) compared to 17β-estradiol [181]. With this, ESR1 dysregulated in different BC subtypes could be regulated by the phytoestrogens present in OT through interacting with estrogen signaling pathways and their chemical structure. Overall, the polyphenols in OT mediate BC tumorigenesis by modulating dysregulated signaling pathways, apoptosis, and hormone receptor signaling, leveraging their pleiotropic interactions to restore cellular homeostasis and disrupt key mechanisms driving tumor progression across BC subtypes.

Here, GO was used to investigate the MOA of the selected bioactive compounds against the DEGs of BC by determining their BP, CC, and MF. The upregulated BC genes showed a share of common annotations regardless of their subtypes. This result suggests that the interaction of the bioactive compounds in OT with the proteins encoded by the hub genes has a similar cellular response to the binding of jasmonic acid. Methyl jasmonate exhibits the previously discussed mechanisms of the anticancer properties of polyphenols in mitigating BC tumorigenesis, which were as follows: arresting the cell cycle, inducing apoptosis and necrosis, disrupting mitochondrial functions, modulating stress signaling, inhibiting proinflammatory enzymes, and suppressing cell migration, angiogenesis, and metastasis [182]. The functional annotation for downregulated genes in BC subtypes, on the contrary, was distinct from each other, which may be due to their characteristic nature of the progression, highlighting their tumor biology, hormonal receptor involvement, molecular signaling pathways, and tumor microenvironment. As discussed, the interaction of OT bioactive compounds with the signaling receptors of downregulated LaBC may suggest enhanced insulin metabolism by stimulating the IGF/insulin signaling pathway. Flavonoids at certain concentrations may mediate the aberrant enzymes responsible for estrogen synthesis and metabolism [183], and the interconnectedness between the abundance of estrogen and insulin may promote the dendritic spine formation through PI3K/Akt/mTOR pathway that could exacerbate BC progression [184]. Furthermore, the enrichment of cellular response to UV-A in LbBC suggests that OT bioactive compounds may exert antioxidant effects. This mitigates oxidative stress in the extracellular matrix (ECM) by attenuating oxidative damage to structural proteins and suppressing NOS activity [185,186], stabilizing the tumor microenvironment and potentially inhibiting angiogenesis and ECM degradation. Moreover, non-luminal BCs (e.g., Her2BC and TNBC) modulated by bioactive compounds in OT have been identified with somewhat similar functional annotations, focusing on cell cycle regulation and cell division through kinase signaling pathways to interfere with BC cells from proliferating, invading, and metastasizing. The collagen catabolic processes and mitotic nuclear membrane disassembly are enriched in the interaction between OT polyphenols and Her2BC and TNBC hub genes, suggesting that it is the target key mechanism in attenuating BC progression. These compounds reinstate the balance between matrix metalloproteinase (MMP) and its endogenous inhibitors to preserve and remodel the extracellular matrix [187,188]. For both BC subtypes, the OT compounds participate in spindle pole and microtubule assembly, which are crucial for maintaining proper chromosome segregation during cell division. The phenolic compounds from OT can modulate mitotic spindle activities by interfering with microtubule polymerization and protecting proper cytoskeleton assembly [189,190]. Moreover, as discussed, the enriched protein serine/threonine/tyrosine kinase activity implicates key signaling pathways, such as PI3K/Akt/mTOR, in the progression of these BC subtypes. OT compounds may inhibit these kinases, reducing cancer cell survival, proliferation, and resistance to apoptosis, thereby addressing oncogenic signaling crucial for tumor growth and metastasis. Thus, the OT bioactive compounds, with their multifaceted interactions and therapeutic versatility, hold transformative potential in BC treatment by not only targeting diverse molecular mechanisms but also restoring cellular homeostasis and impeding tumor progression across subtypes.

### 4.6. Molecular Simulations

To verify the network pharmacology findings, phytochemical OT compounds with favorable pharmacokinetics, pharmacodynamics, and electron-shuttling properties were docked to the proteins encoded by the dysregulated hub genes across BC subtypes, exhibiting great binding affinities in their interaction. The selection of the hub proteins for molecular docking encompassed the common dysregulated hub genes across the upregulated (*PIK3CA* and *PIK3R1*), downregulated (*MAPK1* and *HSP90AB1*), and shared (*IGF1R* and *ESR1*) categories of BC subtypes. Flavonoids demonstrated remarkable binding affinities across multiple BC-related protein targets, particularly those involved in kinase signaling pathways, cell survival and apoptosis regulation, and hormone receptor signaling, suggesting their ability to modulate the stimulation of the hub proteins. Compared to non-flavonoid compounds (e.g., phenolic acids), this enhanced affinity may be attributed to the greater number and positional isomerism of hydroxyl groups in aromatic systems, which likely facilitate stronger hydrogen bonding and other key molecular interactions within the protein binding sites. This, in turn, reduces the intermolecular binding energy and enhances inhibitory properties [191]. However, despite their high affinities, experimental ligands did not consistently outperform standard inhibitors, suggesting that while flavonoids may not serve as direct drug replacements, they could function as complementary dietary agents capable of modulating oncogenic hub proteins to mitigate BC progression. Notably, interactions between estrogen receptors and experimental ligands were an exception, as most flavonoids surpassed tamoxifen in binding affinity. All tested ligands, except catechol, ascorbic acid, and 1-(3,4,5-trihydroxyphenyl)ethenone, demonstrated strong binding affinities to the estrogen receptor, suggesting their potential as alternatives to tamoxifen for hormone receptor-positive BCs. The reduced binding affinity of tamoxifen was likely due to steric hindrance within the receptor’s binding pocket, limiting its interaction with key amino acid residues, whereas the experimental ligands penetrated more deeply and engaged a greater number of residues, resulting in stronger binding affinities. Furthermore, the ligands cyanidin, delphinidin, leucocyanidin, luteolin, quercetin, and tricetinidin not only exhibited strong binding to ESR1 but also demonstrated better interactions with MAPK1 compared to the control drug sorafenib. Their ability to target multiple key regulators of tumorigenesis suggests promising applications in combinatorial or preventive strategies by modulating kinase-driven oncogenic pathways and hormone-responsive tumor phenotypes. Moreover, their natural origin and favorable interaction profiles mark these compounds as promising leads for the development of safer, plant-derived chemopreventive, or possibly anticancer, agents with potentially reduced toxicity compared to conventional chemotherapy.

Luteolin (3′, 4′, 5, 7-tetrahydroxyflavone) emerged as the ligand with the highest binding affinity among all docked compounds across the target hub proteins, prompting further analysis through pharmacophore mapping and molecular dynamics simulations with MAPK and PIK3CA. The abundance of hydroxyl substituents, together with aromatic ring systems, facilitated the stabilization of protein–ligand complexes through non-covalent bonding, including hydrogen bonding, π interactions, and Van der Waals forces. Hydrogen bonding promoted strong binding interactions through the hydrogen bond pairing principle (e.g., strong–strong (s-s) or weak–weak (w-w)) and the release of protein-bound water molecules into the bulk solvent [192,193]. These hydroxyl groups contributed to charge transfer, acting as acceptors in non-polar regions (interaction with valine) and as donors in polar regions on the receptor (interaction with aspartic acid, lysine, tyrosine, and glutamine) [192]. Additionally, the non-polar aromatic rings of luteolin further enhanced its binding affinity to the non-polar amino acid residues on the receptor via Van der Waals interactions, stabilizing the protein structures and complementing the strong affinity with hydrogen bonds. This non-polar aromatic group has an electronegative π-electron cloud system of the aromatic rings formed by the highly conjugated chromone core, the resonance stabilization between the ketone and 5-OH group, and the phenolic hydroxyl groups [194]. In the PIK3CA complex, a π–π T-shaped stacking interaction was observed between the two aromatics rings of tyrosine and luteolin, where the chromone ring of luteolin was positioned perpendicular above the plane of the aromatic ring of tyrosine [195], producing an attractive interaction between their electrons and, thereby, enhancing the binding affinity of luteolin. For both complexes, non-polar amino acid residues on the receptors (e.g., leucine, isoleucine, valine, alanine, and cysteine) interacted with the two benzene rings of luteolin via π–σ and π–alkyl bonding interactions, providing enhanced hydrophobic interactions and stabilized charge transfer for intercalating the ligand with the binding pockets [196]. Moreover, the combination of different non-covalent bonds (i.e., hydrogen bonding, π interactions, and Van der Waals forces) contributed to the overall structural stability and specificity of luteolin binding, highlighting its potential as a lead compound for targeting dysregulated proteins in BC subtypes.

Molecular dynamics simulations revealed that luteolin induced conformational changes in MAPK1 and PIK3CA, driven by the non-covalent bonding interactions of luteolin in its optimal conformational position within the binding pocket, affecting their functions in BC progression. For MAPK1, the molecular dynamics simulations revealed observable conformational changes upon luteolin binding. In contrast to the apo form, which exhibited stable structural deviations throughout the simulation, the holo form displayed noticeable fluctuations, particularly in the initial stages, before stabilizing. These fluctuations indicated that luteolin binding induced structural perturbations in the protein. The changes in RMSF for the holo form compared to the apo form suggest that luteolin binding induces structural rearrangements in MAPK1. Some regions of the protein show reduced flexibility, indicating stabilization, while others exhibit increased flexibility, reflecting a combination of localized stabilization and dynamic adjustment within the binding site. Previous studies have shown that luteolin stabilized MAPK1 protein within a range of ca. 2.30 Å to 6.00 Å over several nanoseconds of simulation [197,198]. Ligand binding influenced the compactness of the protein, as reflected in the Rg, where the holo form demonstrated increased deviation over time, suggesting a more dynamic and expanded conformational state upon ligand binding. Some natural inhibitors of MAPK1 (ZINC02092851, ZINC02130647, and ZINC02133691) and amentoflavone from *Selaginella byropteris* leaf extracts also demonstrated compactness in MAPK1 protein with stable Rg values (~2.16 Å to 2.22 Å) [199,200]. These rearrangements were reflected in the increased SASA in certain regions, suggesting that luteolin-induced conformational changes exposed previously hidden residues to the solvent. This nuanced conformational change, characterized by localized structural rearrangements near the binding pocket, led to solvent exposure to specific regions while maintaining its overall stability [201]. The MMPBSA analysis indicated that the favorable binding of luteolin to MAPK1 is primarily driven by hydrophobic interactions and hydrogen bonding, which stabilize the protein–ligand complex. This suggests an enthalpy-favored binding mechanism, where the favorable enthalpic contributions dominate the overall ΔG. Furthermore, PCA and FEL analyses revealed that the holo form exhibited increased motion in the low-frequency modes, consistent with the observed flexibility. These low-frequency modes correspond to large-scale collective motions in the protein, which are often associated with functional changes. The FEL analysis indicated that luteolin binding stabilized MAPK1 in a local energy minimum, promoting specific structural rearrangements within the binding pocket. Together, the PCA and FEL analyses suggest that luteolin binding not only induces flexibility but also stabilizes MAPK1 in a conformation that facilitates specific functional interactions, which could be crucial for its role in breast cancer progression.

The molecular dynamics simulation in its apo and holo protein states revealed notable differences in conformational stability and flexibility upon luteolin binding, suggesting a potential regulatory role. The RMSD analysis demonstrated that the holo protein underwent a continuous conformational shift, stabilizing at a higher deviation compared to the apo state, indicating that luteolin binding induced structural changes without de-stabilizing the protein. The RMSF analysis further supported this, with luteolin binding induced fluctuating conformational changes, having an increased flexibility in some regions and stabilization in others. At the binding site, both enhanced and reduced RMSF were observed, indicating the role of luteolin in modulating the flexibility and rigidity of the protein. The Rg analysis further highlighted the dynamic differences, with the holo protein showing increasing compactness, while the apo protein fluctuated more, pointing to a potential increase in structural stability upon luteolin binding. SASA measurements indicated that the holo state had a slightly more stable exposure to the solvent, suggesting enhanced accessibility for regulatory interactions, whereas the apo state had greater sol-vent exposure overall. ΔG calculations confirmed the stability of the complex, characterized by a negative enthalpy change and a smaller opposing entropy term, indicating an enthalpy-favored binding mechanism dominated by favorable non-covalent interactions. Additionally, PCA/FEL analysis showed significant shifts in the energy landscape, indicating the dynamic nature of the holo-PIK3CA complex over time. These findings suggest that luteolin modulates the structural and dynamic properties of PIK3CA, potentially influencing its role in cellular signaling pathways such as PI3K/AKT/mTOR, which are critical for cancer progression.

Overall, the polyphenol-rich OT has been shown to mitigate oxidative stress caused by the imbalance of free radicals over the antioxidant, thereby alleviating the acidity of the tumor microenvironment—a factor known to alter key biological mechanisms in BC development. This protective effect lies in the chemical structure of its bioactive compounds, specifically antioxidants and electron shuttles, which were hydroxyl groups and benzene rings with *ortho*- and *para*-dihydroxyl substituents, that provide an optimal electron transfer between the electron donors and acceptors. Electrochemical analyses revealed that the OT water extracts, specifically OTL, yielded a substantial amount of electron shuttles, while ethanolic OT extracts favored the yield of the antioxidants. However, the OTL-E extracted phytochemicals at levels that became toxic to electroactive bacteria, affecting their bioenergy-stimulating properties. Furthermore, the hydroxyl groups in the OT bioactive compounds deprotonate as the pH level increases, which is reflected in the attenuation of redox potential curves. In relation to the MOA against BC, the OT compounds exhibited chemopreventive properties in BC development by targeting dysregulated proteins involved in kinase signaling, cell survival, apoptosis regulation, and hormone receptor pathways. Notably, flavonoids in OT acted as reversible catalysts that modulated key hub proteins implicated in BC tumorigenesis, including ESR1, PIK3CA, and MAPK1, with exceptional and spontaneous binding interaction. Particularly, the luteolin binding to PIK3CA and MAPK1 was attributed to the combination of non-covalent interactions that enthalpy-favored conformations in these proteins, potentially altering their activity upon interaction with their respective agonists. These findings suggest that the bioactivity of the natural products in OT may play a crucial role in disrupting key oncogenic pathways, offering a promising avenue for BC prevention and treatment. Our current study has contributed to growing information on the health benefits of functional foods, including Oolong tea, on cancer.

## 5. Conclusions

This in vitro and in silico study explored the medicinal potential of OT extracts, obtained through various extraction techniques and solvents, in BC treatment by integrating phytochemical analysis, antioxidant analysis, electrochemical profiling, network pharmacology, and molecular docking and dynamics simulations. While OTL-E demonstrated remarkable phytochemical and antioxidant properties, OTL-W exhibited better electron-mediating capabilities, as validated through MFCs at 2000 ppm and CV at 50 cycles in neutral pH, demonstrating its potential role in bioenergy production and electron transport for ATP synthesis. SE with water as solvent emerged as the optimal method for yielding ESs in OT compared to OTS extracts, suggesting that OT extracts rich in electron mediators could serve as a promising herbal approach for BC prevention and treatment. Network pharmacology analysis further corroborated the therapeutic potential of the bioactive compounds in OT, demonstrating their ability to modulate dysregulated proteins involved in kinase signaling, cell survival, apoptosis regulation, and hormone receptor pathways for alleviating BC development. Flavonoids, acting as reversible catalysts, exhibited spontaneous binding with key dysregulated oncogenic proteins, achieving binding affinities comparable to control inhibitors, highlighting their modulatory potential in BC tumorigenesis. Notably, luteolin binding was driven by non-covalent interactions with residues of hub proteins, inducing conformational changes in downregulated MAPK1 and upregulated PIK3CA, thereby confirming the therapeutic properties of OT. This comprehensive investigation underscores OT as a multifunctional therapeutic candidate, bridging antioxidant, electrochemical, and pharmacological mechanisms for potential BC intervention. Regular consumption of Oolong tea may enhance its protective effects by maximizing its bioactive potential, supporting BC prevention and treatment. To advance OT-based therapies, future studies could employ systems biology approaches, such as metabolomics and proteomics, to elucidate the holistic effects of OT on BC cancer patients in vivo by integrating multi-omics data, alongside liquid chromatography-mass spectrophotometry to identify predominant compounds and their mechanistic roles in disease modulation.

## Figures and Tables

**Figure 1 biology-14-00487-f001:**
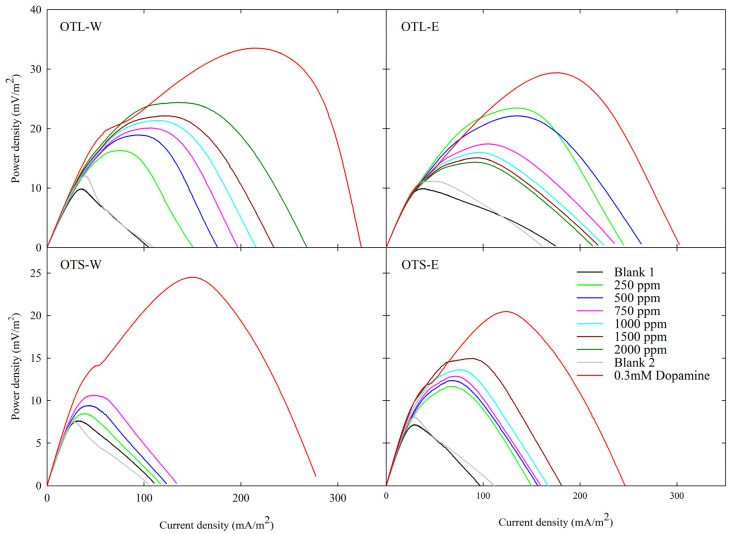
Power densities of different extraction methods of OT using MFC as a bioenergy evaluating platform.

**Figure 2 biology-14-00487-f002:**
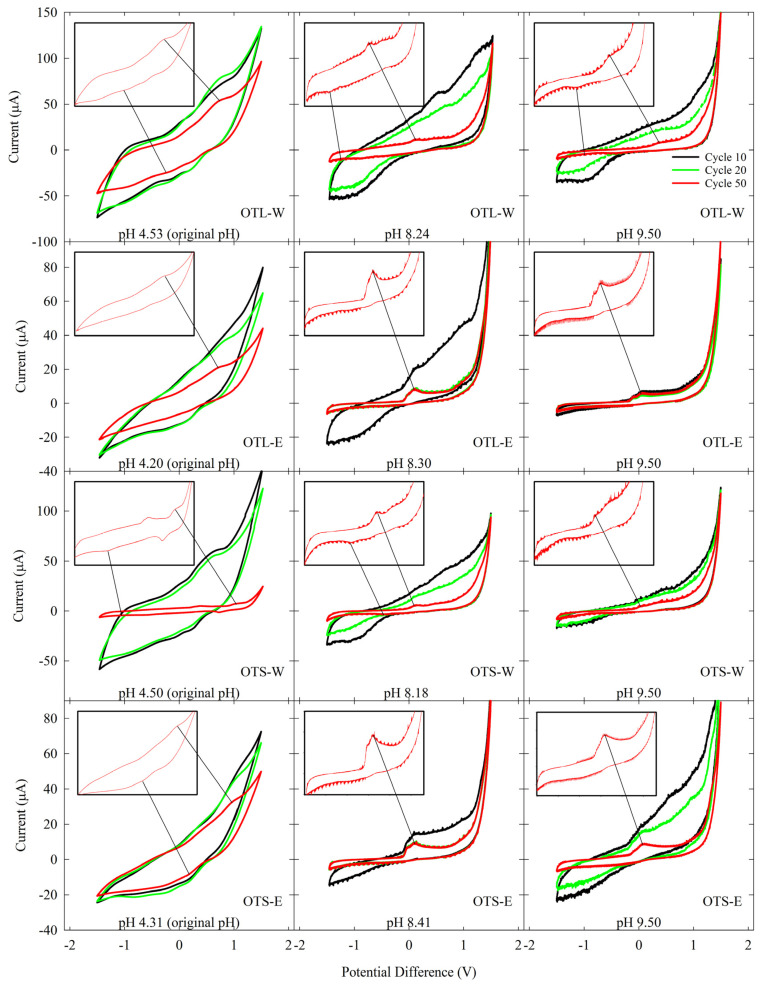
Comparative cyclic voltammograms for cycles 10 (black), 20 (green), and 50 (red) of OT extracts using carbon black-modified glassy carbon working electrode, performed in varying extraction methods, solvent systems, and pH levels (i.e., original pH of the extracts, pH 8.5, pH 10) at 25 °C. The significant reduction and oxidation potential peaks are 50 cycles were enlarged and emphasized within the graph.

**Figure 3 biology-14-00487-f003:**
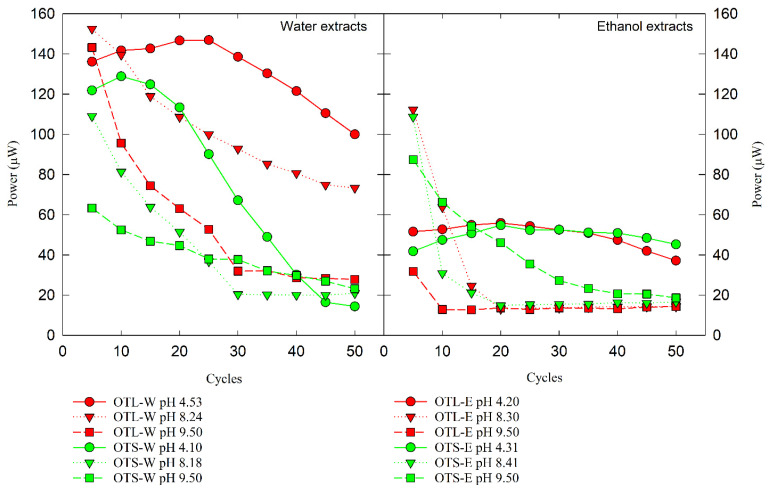
Comparative CV area as power generated of OTL and OTS extracts with water (W) and ethanol (E) as solvents in varying pH levels (original, 8.5, and 10).

**Figure 4 biology-14-00487-f004:**
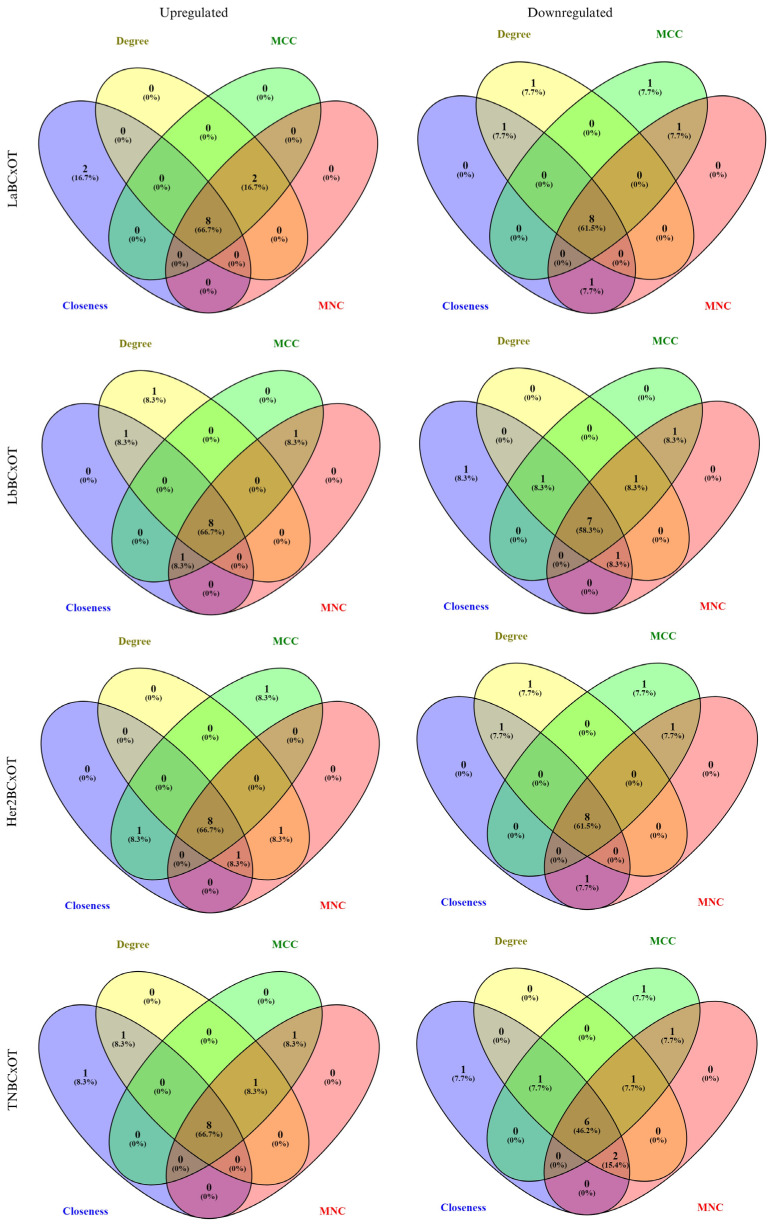
Identification of hub genes in BC subtypes. Venn diagrams illustrate the overlap of hub genes identified using four centrality measures: degree, MCC, closeness, and MNC. The analysis was performed on downregulated (top row) and upregulated (bottom row) gene sets for four BC subtypes: LaBCxOT, LbBCxOT, Her2BCxOT, and TNBCxOT. Numbers within overlapping regions indicate the number of genes shared by the corresponding centrality measures.

**Figure 5 biology-14-00487-f005:**
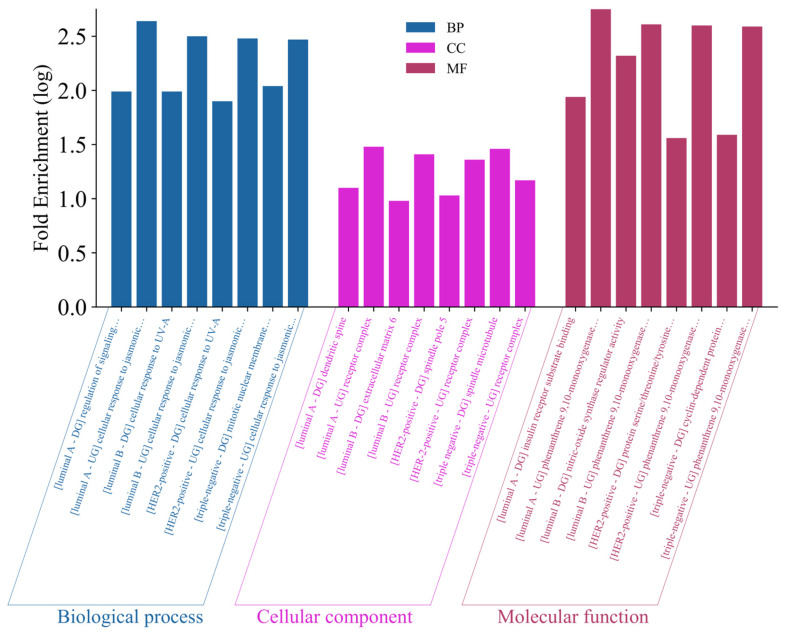
GO enrichment analysis reveals distinct functional profiles in BC subtypes. The bar chart displays the significantly enriched GO terms for upregulated (UG) and downregulated (DG) gene sets across different BC subtypes. Enrichment scores are represented as log-transformed fold enrichment and are categorized by GO domains: BP, CC, MF.

**Figure 6 biology-14-00487-f006:**
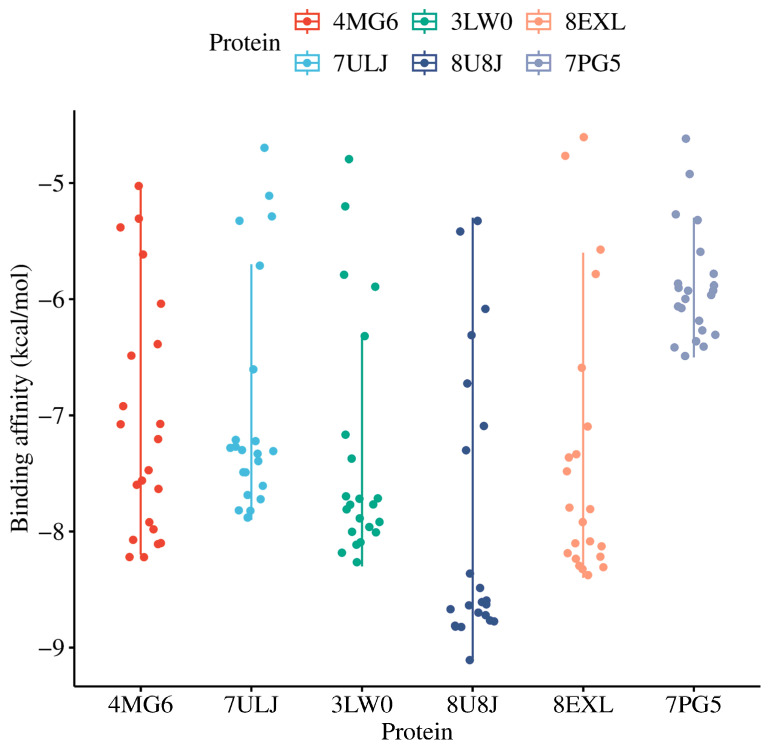
Binding affinities of electron-shuttling compounds present in OT and controls against different proteins encoded by the hub genes of BC. Binding affinities are expressed in kcal/mol.

**Figure 7 biology-14-00487-f007:**
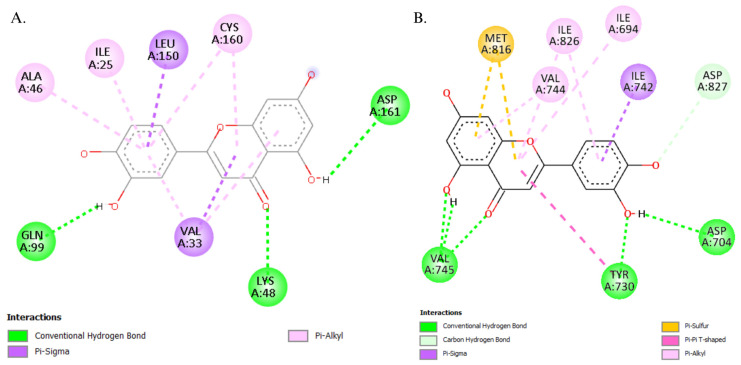
Binding interaction of luteolin to the amino acid residues of the key hub proteins across the deregulated genes of BC subtypes: (**A**) MAPK1 for downregulated and (**B**) PIK3CA for upregulated genes.

**Figure 8 biology-14-00487-f008:**
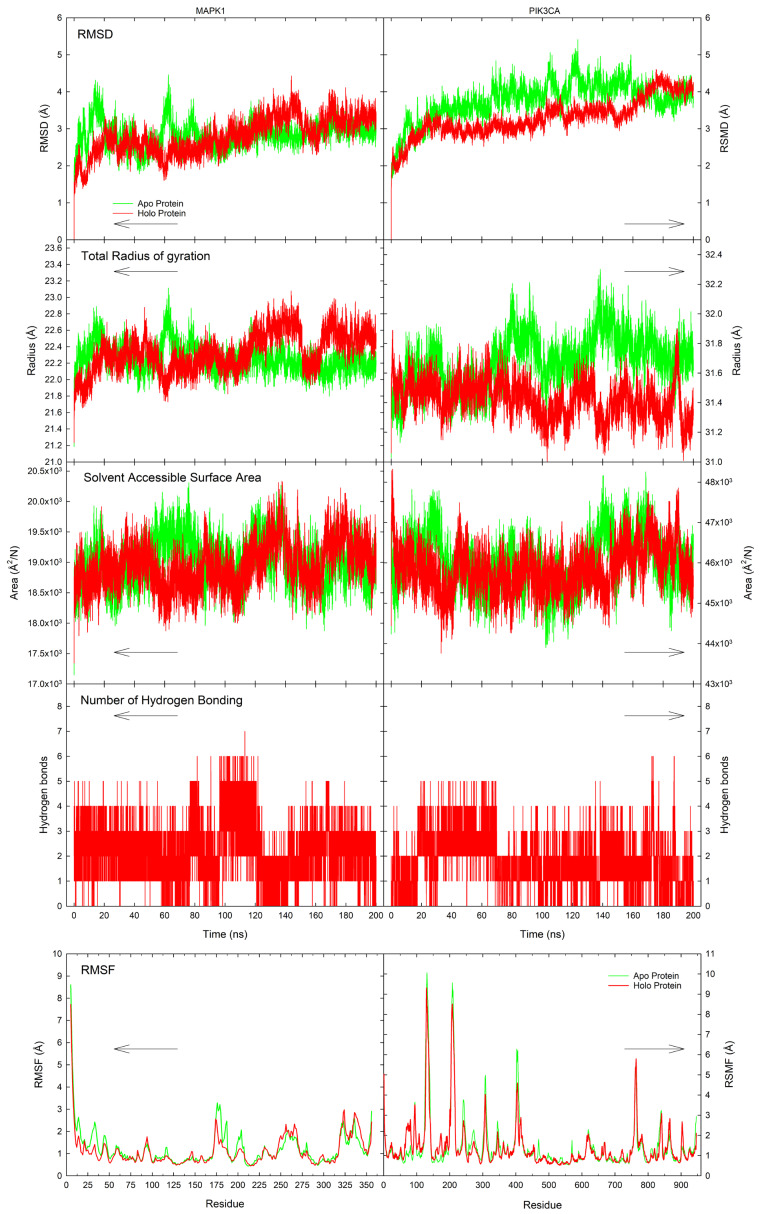
A 200 ns molecular dynamics simulation illustrating the protein–ligand interactions between luteolin and the protein structures of MAPK1 (**left**) and PIK3CA (**right**), comparing their apo (green) and holo (red) states.

**Figure 9 biology-14-00487-f009:**
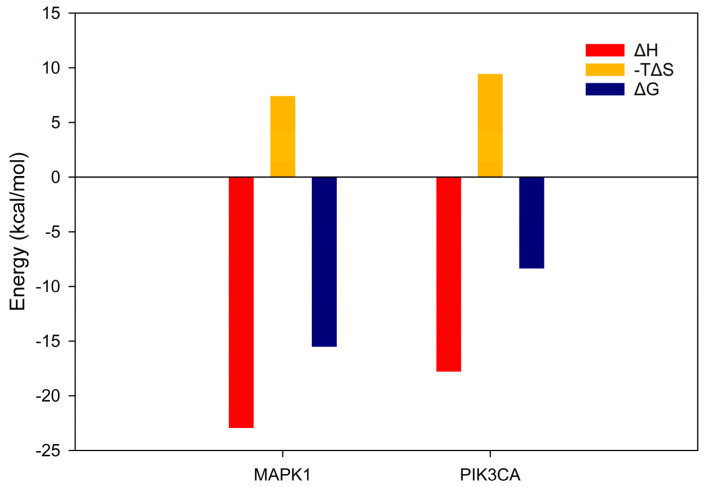
MMPBSA binding free–energy decomposition for the MAPK1 and PIK3CA complexes with luteolin, calculated for the time windows 50–70 ns (MAPK1) and 100–120 ns (PIK3CA). The total enthalpy (ΔH; red), entropy (-TΔS; yellow), and free–energy (ΔG; blue) values are shown.

**Figure 10 biology-14-00487-f010:**
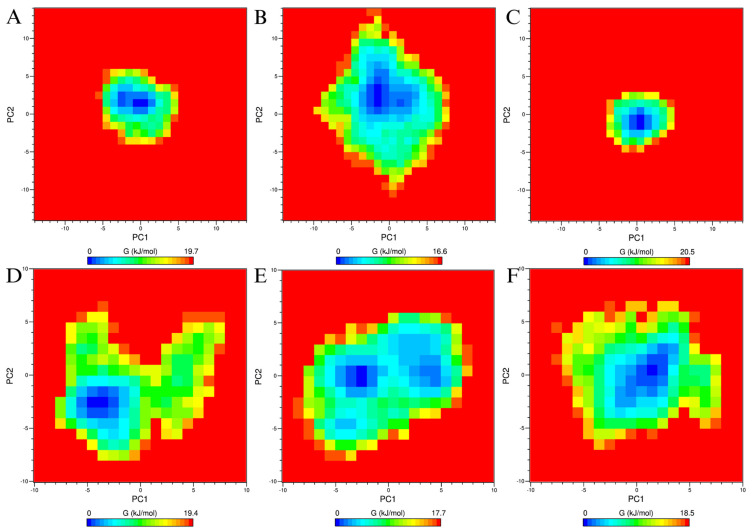
PCA/FEL analysis of the MAPK1–luteolin (top row) and PIK3CA–luteolin (bottom row) complexes across distinct simulation time windows. (**A**–**C**) correspond to MAPK1 at 0–20 ns, 50–70 ns, and 180–200 ns, respectively. (**D**–**F**) correspond to PIK3CA at 0–20 ns, 100–120 ns, and 180–200 ns, respectively. The color gradient indicates the ΔG (in kJ/mol), with red denoting areas of high ΔG and blue indicating regions of low ΔG. The units of Principal Components 1 (PC1) and 2 (PC2) are expressed in nm.

**Table 1 biology-14-00487-t001:** Predicted binding sites of the hub proteins using PrankWeb.

Hub Gene	PDB ID	Coordinates			Resolution	Ref
		X	Y	Z		
*ESR1*	4MG6	22.3896	5.54	5.3394	2.10 Å	[67]
*HSP90AB1*	7ULJ	−22.3339	100.8248	4.0096	1.82 Å	[68]
*IGF1R*	3LW0	−5.6064	−22.6792	−53.4945	1.79 Å	[69]
*MAPK1*	8U8J	10.5209	14.9872	16.18	2.10 Å	[70]
*PIK3CA*	8EXL	−18.0216	13.4051	27.3344	1.99 Å	[71]
*PIK3R1*	7PG5	−56.7456	−21.2754	43.3115	2.20 Å	[72]

**Table 2 biology-14-00487-t002:** Phytochemical content analysis of OT extracts.

Extract	TPC Analysis (GAE mg/g)	TFC Analysis (RE mg/g)	TCTC Analysis (CE mg/g)
OTL-E	352.226 ± 0.017	72.6407 ± 0.005	428.884 ± 0.012
OTL-W	249.497 ± 0.005	34.6878 ± 0.002	9.39799 ± 0.0002
OTS-W	273.433 ± 0.006	33.6675 ± 0.001	304.518 ± 0.001

**Table 3 biology-14-00487-t003:** Antioxidant activity of OT extract.

Sample	DPPH IC_50_ (mg/mL)	FRAP (Trolox mg/g)
OTL-E	0.07282 ± 0.006	177.604 ± 0.590
OTL-W	0.10907 ± 0.006	142.21 ± 0.957
OTS-W	0.099131 ± 0.002	150.14 ± 1.978
Ascorbic Acid	0.0591 ± 0.002	

**Table 4 biology-14-00487-t004:** Comparison of PD generation (Mean ± SD) across various OT extracts and concentrations, with PD amplification with respect to blank shown as subscript. PD values are expressed in mV/m^2^.

Test Sample	OTL-W	OTL-E	OTS-W	OTS-E
Blank 1	9.882 ± 0.494_1.00±0.10_	11.193 ± 0.798_1.00±0.1411_	7.604 ± 0.471_1.00±0.12_	7.150 ± 0.365_1.00±0.10_
250 ppm	16.349 ± 1.138_1.65±0.20_	23.500 ± 4.757_2.10±0.57_	8.477 ± 1.509_1.11±0.27_	11.636 ± 1.509_1.63±0.29_
500 ppm	18.946 ± 1.741_1.92±0.27_	22.158 ± 4.717_1.98±0.56_	9.404 ± 1.414_1.24±0.26_	12.363 ± 1.414_1.73±0.29_
750 ppm	20.129 ± 1.507_2.04±0.25_	17.459 ± 2.446_1.56±0.33_	10.617 ± 1.426_1.40±0.27_	12.874 ± 1.426_1.80±0.29_
1000 ppm	21.369 ± 1.875_2.16±0.30_	16.001 ± 2.385_1.43±0.31_	-	13.614 ± 1.475_1.90±0.30_
1500 ppm	22.166 ± 2.026_2.24±0.32_	15.091 ± 2.024_1.35±0.28_	-	14.961 ± 1.446_2.09±0.31_
2000 ppm	24.374 ± 4.113_2.47±0.54_	14.376 ± 1.963_1.28±0.27_	-	-
Blank 2	12.070 ± 0.506_1.22±0.11_	9.929 ± 0.762_0.89±0.13_	7.488 ± 0.309_0.98±0.10_	8.051 ± 0.430_1.13±0.12_
Dopamine	33.531 ± 2.089_3.39±0.38_	29.407 ± 2.975_2.63±0.45_	24.496 ± 2.323_3.22±0.51_	20.473 ± 1.661_2.86±0.38_

## Data Availability

The original data presented in the study are openly available in GEO-NCBI in https://doi.org/10.15252/emmm.201505891 or GSE45827.

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
