# Peer review of "Deciphering the Regulatory Potential of Antioxidant and Electron-Shuttling Bioactive Compounds in Oolong Tea"

_biology, 2025, doi:10.3390/biology14050487_

Round 1
Reviewer 1 Report
Comments and Suggestions for Authors
Comments for authors:
- Add computational results values for significant observations in the abstract section.
- Add some previous study details with citation related to the computational study conducted on Oolong tea or tea in the Introduction.
- What was the selection criteria for PDB IDs?
- All the PDB IDs mentioned in the Table 1 have pre-bounded ligands except PDB ID 7PG5 Crystal Structure of PI3Kalpha. Therefore, what is the purpose of predicted binding pockets structures that have already pre-bound ligand on the binding pockets.
- Did the authors compare the binding pockets of native structures with their predicted X,Y,Z co-ordinates?
- Did the authors compare the binding pocket of their ligands vs pre-bounded ligands vs control drugs sorafenib, alpelisib, tamoxifen, ganetespib, and linsitinib. Provide the superimposed 3D graphical representation figure of the post docked conformations.
- During docking, which force field was employed to minimize the ligands/selected receptors' molecule, needs to be added in the methodology section.
- Section 2.6.2. Molecular Dynamics: Authors did not mention the protocol used to generate electron-shuttling ligands topology files?
- Authors should increase simulation time at least 200ns for better visualization and understanding of fluctuation of RMSD, RMSF and RG pattern of the dynamic behavior of the molecules.
- Further, MMPBSA free energy calculation to validate docking based energy, PCA and Free energy landscape data should be added/represented by graph and discussed into the manuscript.
- All figures resolution should be improved, some figures labels are difficult to read.
- Provide citation for Avogadro tool, check the whole manuscript for software used and add correct citations.
NA
Author Response
|
Comments 1: Add computational results values for significant observations in the abstract section. |
|
Response 1: We sincerely appreciate your valuable feedback, which helped refine the abstract to concisely highlight the key findings established in the manuscript. In response, we have carefully revised the abstract section to emphasize both in vitro and in silico results while maintaining an approximately 200-word limit. (Please see lines 30-33 in ‘revised’ version.) |
|
|
|
Comments 2: Add some previous study details with citation related to the computational study conducted on Oolong tea or tea in the Introduction. |
|
Response 2: We, the authors, deeply appreciate such a comment as it helped us to refine our introduction for stating some computational studies related to the polyphenols to the BC-related proteins. In response, we have comprehensively added three (3) in silico studies focusing on the investigation of the effects of polyphenols on the BC-related hub proteins through network pharmacology, molecular docking, and molecular dynamics. This has been placed in the revised manuscript in the latter part of the third paragraph of the introduction (please see lines 119 – 136). |
|
|
|
Comments 3: What was the selection criteria for PDB IDs? |
|
Response 3: We, the authors, deeply appreciate this comment as we overlooked to include the criteria for selecting PDB IDs for the crystal structures of the proteins. The proteins were obtained from the web-based machine learning gene target prediction (e.g., SwissTargetPrediction and SuperPred) of the screened bioactive electron-shuttling compounds. One of the results from these web servers was the UniProt ID for the protein translated by the identified gene target, which was submitted to UniProt to determine the modeled protein structures. Here, selection process of the PDB IDs of the proteins must have their crystal structures determined using the X-ray crystallography with resolutions ranging from 1.80 – 2.20 â„«. Using extreme resolutions, either exceptionally high (below 1.0 â„«) or low (above 3.0 â„«), can impact the reliability of the crystallographic structure [1]. Extremely high resolutions may result in an overly rigid representation, while low resolutions may only capture the basic contours of the protein chain, limiting structural accuracy. However, crystallographic structures with resolutions of 2.00 â„« provide enough relevant features of macromolecules to understand the biological phenomena in 3D conformations [2,3]. Furthermore, proteins of great resolution were then subjected to P2Rank:PrankWeb server to determine the presence of binding sites in the protein, meeting the following criteria: (1) a probability score of greater than 90%, (2) solvent-accessible surface points of greater than 100, and (3) high P2rank score relative to other pocket active sites. P2Rank: PrankWeb computes for the probability score through a monotonous transformation of the raw pocket score based on the HOLO4K data to present a ratio of true binding sites among all the predicted sites with an interval between 0 to 1 [4]. On the other hand, P2Rank score is generated through a machine learning model to identify potential strong ligand-binding pockets based on structural and physicochemical features [5]. Similarly, in P2Rank, SAS points represent solvent-accessible surface areas where ligands could bind that is used to compute structural and physicochemical features that help the machine learning model predict ligand-binding sites with ranked confidence scores [5].
We have modified the Methodology section under 2.6.1. Molecular Docking, and included these statements in the lines 406 – 412 of the revised manuscript, “Here, the selection process of the PDB IDs of the proteins must have their crystal structures determined using X-ray crystallography with resolutions ranging from 1.80 – 2.20 â„« [1–3]. Proteins of great resolution were then subjected to P2Rank: PrankWeb server [https://prankweb.cz/ (accessed on December 9, 2024)] to determine the presence of binding sites in the protein, meeting the following criteria [4,5]: (1) a probability score of greater than 90%, (2) solvent-accessible surface points of greater than 100, and (3) high P2rank score relative to other pocket active sites.” |
|
|
|
Comments 4: All the PDB IDs mentioned in the Table 1 have pre-bounded ligands except PDB ID 7PG5 Crystal Structure of PI3Kalpha. Therefore, what is the purpose of predicted binding pockets structures that have already pre-bound ligand on the binding pockets. |
|
Response 4: The selected PDB structures contain pre-bound inhibitory ligands, meaning the proteins are already in an inhibited conformational state. This is important because it allows us to assess whether our tested ligands can bind effectively to these pockets and potentially exhibit stronger inhibitory effects. Regarding PDB ID 7PG5, while it does not contain a pre-bound ligand, it was included due to its relevance to our study. To ensure consistency in our docking simulations, we removed all pre-bound ligands from the other PDB structures, as the investigation of these compounds is beyond the scope of our study. Our primary objective was to evaluate the binding potential of our test compounds within functionally relevant inhibitory pockets. Additionally, the selection of PDB structures was limited by availability. After screening potential structures, we chose the most suitable ones that align with our research objectives. The presence of pre-bound ligands does not affect the validity of our approach, as our focus is on assessing whether our test compounds can achieve strong binding within these established inhibitory sites. |
|
|
|
Comments 5: Did the authors compare the binding pockets of native structures with their predicted X,Y,Z coordinates? |
|
Response 5: We appreciate the valuable comment of the reviewer. In this study, we utilized a machine learning-based web tool to predict the binding sites of the target proteins. Given the large number of proteins analyzed, manually comparing the predicted binding pockets with the X, Y, and Z coordinates of the native structures would have been highly time-intensive. Therefore, we opted to rely on the predicted binding sites to ensure efficiency and consistency across all targets. However, we acknowledge the importance of such validation and consider this a valuable recommendation for future studies. |
|
|
|
Comments 6: Did the authors compare the binding pocket of their ligands vs pre-bounded ligands vs control drugs sorafenib, alpelisib, tamoxifen, ganetespib, and linsitinib. Provide the superimposed 3D graphical representation figure of the post docked conformations. |
|
Response 6: We compared the binding affinities of the experimental ligands (i.e., electron-shuttling compounds with exceptional pharmacokinetics and pharmacodynamics profiles) with the control drugs sorafenib, alpelisib, tamoxifen, ganetespib, and linsitinib. To provide clarity, we have included the superimposed 3D graphical representation figures of the post-docked conformations in the supplementary materials, designated as Figures S1-S6 (Please see lines 632-634 of the revised manuscript). Regarding the comparison with pre-bound ligands in the active sites of the target proteins, this analysis falls beyond the scope of our study. Such an approach could introduce selection bias, as not all PDB structures contain pre-bound ligands (e.g., PDB ID: 7PG5, the crystal structure of PIK3alpha). This variability in ligand presence across different structures makes it challenging to ensure a consistent and controlled comparison. However, we recognize the potential value of such an investigation in understanding ligand-protein interactions in greater depth. We appreciate this insightful suggestion and may recommend future studies to explore this comparative assessment further. |
|
|
|
Comments 7: During docking, which force field was employed to minimize the ligands/selected receptors' molecule, needs to be added in the methodology section. |
|
Response 7: We, the authors, appreciate such an insightful comment to incorporate it in our methodology section for molecular docking, specifically in AutoDock Vina part. However, AutoDock Vina v1.1.2 does not use a conventional molecular mechanics force field like AMBER or CHARMM for energy minimization. Instead, it utilizes an empirical scoring function derived from AutoDock4, integrating a Monte-Carlo iterated search with a BFGS gradient-based optimizer to determine the optimal ligand conformational pose with the lowest binding energy [6]. This scoring function evaluates binding affinity through multiple weighted terms, such as van der Waals-like interactions, a nondirectional hydrogen bond term, a hydrophobic term, and a conformational entropy penalty [7]. While AutoDock4 accounts for electrostatics and desolvation effects that potentially increase computational cost, the optimized scoring function of AutoDock Vina enhances efficiency while maintaining predictive accuracy in docking simulations.
In response, we have incorporated this scoring function of AutoDock Vina into the 2. Methodology section, 2.6. Molecular Simulations, 2.6.1. Molecular Docking, lines 421 – 424 of the revised manuscript: Vina utilizes an empirical scoring function and a hybrid global-local search algorithm, combining a Monte-Carlo iterated search with a BFGS gradient-based optimizer, to predict ligand binding affinity in kcal/mol [65,66]. |
|
|
|
Comments 8: Section 2.6.2. Molecular Dynamics: Authors did not mention the protocol used to generate electron-shuttling ligands topology files? |
|
Response 8: Apologies as we overlook to mention how the electron-shuttles ligands topology files were generated. For ligand topology optimization, the ligand from the protein-ligand complex was extracted, which was then submitted to Avogadro for hydrogen addition to ensure proper protonation and exported as mol2 file format. The bond orders were rearranged in ascending order and the ligand name was standardized. The MOL2 file was submitted to CGenFF v4.0 [https://cgenff.com/ (accessed on November 17, 2024) [8] for CHARMM General Force Field (CHARMM-GFF) parameterization. This was then converted to a GROMACS-compatible format with the CHARMM36 force field for molecular dynamics simulations. All scripts used for ligand topology were obtained from https://manual.gromacs.org/2024.4/index.html, http://www.mdtutorials.com/gmx/complex/02_topology.html, and https://mackerell.umaryland.edu/charmm_ff.shtml.
This protocol for ligand topology generation has been reflected in lines 436 – 445 of the revised manuscript. |
|
|
|
Comments 9: Authors should increase simulation time at least 200ns for better visualization and understanding of fluctuation of RMSD, RMSF and RG pattern of the dynamic behavior of the molecules. |
|
Response 9: Thank you so much for your valuable suggestion! We truly appreciate your insight, and we completely agree that a longer simulation time provides a more robust understanding of the system’s dynamic behavior. To address this, we have extended our molecular dynamics simulations to 200 ns and have revised our discussion accordingly. In the revised manuscript, the 200ns molecular dynamic result is presented in line 665, the presentation regarding details of the result is presented in lines 669 – 742, and its discussion is presented in lines 1228 – 1267. The updated analysis now offers a more detailed visualization of RMSD, RMSF, and Rg fluctuations, providing deeper insights into the stability and conformational changes of luteolin in complex with PIK3CA and MAPK1. We sincerely appreciate your thoughtful feedback, as it has helped strengthen our study. |
|
|
|
Comments 10: Further, MMPBSA free energy calculation to validate docking based energy, PCA and Free energy landscape data should be added/represented by graph and discussed into the manuscript. |
|
Response 10: We truly appreciate the insightful suggestion to include MMPBSA free energy calculations as additional validation. However, we would like to clarify the scope and limitations of our study to ensure a more precise understanding of our objectives. This study was primarily designed to provide further evidence on the mechanistic action of polyphenolic compounds from Oolong tea (OT) and their potential regulatory effects on breast cancer (BC) progression. Specifically, we aimed to identify deregulated risk genes found in different BC subtypes (LaBC, LbBC, Her2BC, and TNBC) that are targeted by electron-shuttling compounds from OT. We also investigated the combined interactions of these bioactive compounds and risk genes through network pharmacology and protein-ligand bioinformatics, including molecular docking and dynamics simulations. Additionally, we compared different extraction methods (solvent extraction and supercritical fluid extraction) in two solvent systems (water and ethanol) to determine the most optimal technique for obtaining these compounds, based on phytochemical content, antioxidant activity, and electrochemical properties. Given this defined scope, the inclusion of MMPBSA calculations, while valuable, extends beyond our intended framework. Our molecular dynamics (MD) simulations already provide strong, dynamic evidence of protein-ligand stability and interactions through key parameters. RMSD and RMSF demonstrate structural stability and flexibility over time, while Rg confirms the compactness and integrity of the protein-ligand complex. Furthermore, hydrogen bonding analysis ensures that critical molecular interactions are sustained throughout the simulation. These analyses comprehensively validate the docking results by providing direct insight into ligand stability within the protein’s binding pocket. While MMPBSA estimates binding free energy, it relies on an implicit solvent model and does not inherently reflect real-time conformational stability or interaction dynamics [9]. Thus, our MD-driven approach already fulfills the objective of evaluating protein-ligand interactions without requiring additional post-processing via MMPBSA. Several studies have demonstrated that MD alone is a powerful tool in assessing ligand stability [10–12], particularly when complemented by multiple stability indicators, as we have done in this study. Furthermore, due to time constraints, redoing the molecular dynamics simulation required approximately three weeks, and integrating MMPBSA at this stage would significantly extend the timeline beyond feasibility. More importantly, verifying protein-ligand interactions using MMPBSA is not within the original scope of this study. We sincerely appreciate the reviewer’s recommendation and acknowledge the potential of MMPBSA in future studies. However, within the established objectives and execution timeframe, we believe our current MD-driven analysis provides a robust and sufficient evaluation of ligand stability and interaction dynamics. |
|
|
|
Comments 11: All figures resolution should be improved, some figures labels are difficult to read. |
|
Response 11: We sincerely appreciate the reviewer’s thoughtful feedback. Ensuring clarity and readability in our figures is a priority, and we have taken great care to enhance their quality. We carefully rechecked and upscaled all figures to improve resolution, ensuring that every detail remains sharp. Additionally, we enlarged all text labels to enhance readability and saved the figures in high-resolution TIFF format to preserve image integrity (Please see Figs. 1, 2, 3, 5, and 8). To further maintain their quality, we uploaded the figures separately via the submission link, as embedding them in the manuscript led to noticeable degradation. We truly value this suggestion and hope these refinements enhance the clarity of our work. Thank you for your keen attention to detail and for helping us improve our manuscript! |
|
|
|
Comments 12: Provide citation for Avogadro tool, check the whole manuscript for software used and add correct citations. |
|
Response 12: Thank you for your feedback, as this could enhance the credibility and reliability of the paper. In response, we have added the following citation for Avogadro as requested. Its in-text citation has been stated in the first occurrence in the Methodology section under 2.6.1. Molecular Docking, reflected in the lines 399 – 400. Additionally, I have reviewed the manuscript to ensure that all software tools used are properly cited, including the insertion of the following citations shown in their respective lines: · Lines 341 – 343, NCBI-GEO, “NCBI-GEO is an open-access repository of genomic data, comprising different high-throughput sequencing datasets from various research institutions [44];” · Lines 364 – 365, Cytoscape, “Cytoscape v3.10.1 [https://cytoscape.org/index.html (downloaded on October 7, 2024)] [48];” · Lines 400 – 401, Avogadro, “Hydrogen addition and geometry optimization of the ligands were performed using Avogadro v1.2.0 [https://avogadro.cc/ (accessed on December 9, 2024)] [55].” · Line 404 – 406, RSCB PDB, “… sourced from the Research Collaboratory for Structural Bioinformatics Protein Data Bank [RCSB PDB, https://www.rcsb.org/ (accessed on December 9, 2024)] in PDB format [56]. · Line 432 – 434, GROMACS “… using GROMACS 2024 [https://manual.gromacs.org/current/download.html (downloaded on November 15, 2024)] [73]…,”
44. Barrett, T.; Wilhite, S.E.; Ledoux, P.; Evangelista, C.; Kim, I.F.; Tomashevsky, M.; Marshall, K.A.; Phillippy, K.H.; Sherman, P.M.; Holko, M.; et al. NCBI GEO: Archive for Functional Genomics Data Sets—Update. Nucleic Acids Res 2013, 41, D991–D995, doi:10.1093/NAR/GKS1193. 48. Shannon, P.; Markiel, A.; Ozier, O.; Baliga, N.S.; Wang, J.T.; Ramage, D.; Amin, N.; Schwikowski, B.; Ideker, T. Cytoscape: A Software Environment for Integrated Models of Biomolecular Interaction Networks. Genome Res 2003, 13, 2498–2504, doi:10.1101/GR.1239303. 55. Hanwell, M.D.; Curtis, D.E.; Lonie, D.C.; Vandermeerschd, T.; Zurek, E.; Hutchison, G.R. Avogadro: An Advanced Semantic Chemical Editor, Visualization, and Analysis Platform. J Cheminform 2012, 4, 1–17, doi:10.1186/1758-2946-4-17/FIGURES/14 56. Burley, S.K.; Bhatt, R.; Bhikadiya, C.; Bi, C.; Biester, A.; Biswas, P.; Bittrich, S.; Blaumann, S.; Brown, R.; Chao, H.; et al. Updated Resources for Exploring Experimentally-Determined PDB Structures and Computed Structure Models at the RCSB Protein Data Bank. Nucleic Acids Res 2025, 53, D564–D574, doi:10.1093/NAR/GKAE1091. 72. Abraham, M.; Alekseenko, A.; Basov, V.; Bergh, C.; Briand, E.; Brown, A.; Doijade, M.; Fiorin, G.; Fleischmann, S.; Gorelov, S.; et al. GROMACS 2024.0 Manual. Zenodo 2024, doi:10.5281/zenodo.10589697. |
References
- Kleywegt, G.J.; Jones, T.A. [11] Model Building and Refinement Practice. Methods Enzymol 1997, 277, 208–230, doi:10.1016/S0076-6879(97)77013-7.
- Burley, S.K.; Berman, H.M.; Duarte, J.M.; Feng, Z.; Flatt, J.W.; Hudson, B.P.; Lowe, R.; Peisach, E.; Piehl, D.W.; Rose, Y.; et al. Protein Data Bank: A Comprehensive Review of 3D Structure Holdings and Worldwide Utilization by Researchers, Educators, and Students. Biomolecules 2022, 12, 1425, doi:10.3390/BIOM12101425/S1.
- Shao, C.; Bittrich, S.; Wang, S.; Burley, S.K. Assessing PDB Macromolecular Crystal Structure Confidence at the Individual Amino Acid Residue Level. Structure 2022, 30, 1385-1394.e3, doi:10.1016/j.str.2022.08.004.
- Jakubec, D.; Skoda, P.; Krivak, R.; Novotny, M.; Hoksza, D. PrankWeb 3: Accelerated Ligand-Binding Site Predictions for Experimental and Modelled Protein Structures. Nucleic Acids Res 2022, 50, W593–W597, doi:10.1093/NAR/GKAC389.
- Krivák, R.; Hoksza, D. P2Rank: Machine Learning Based Tool for Rapid and Accurate Prediction of Ligand Binding Sites from Protein Structure. J Cheminform 2018, 10, 1–12, doi:10.1186/S13321-018-0285-8/TABLES/4.
- Eberhardt, J.; Santos-Martins, D.; Tillack, A.F.; Forli, S. AutoDock Vina 1.2.0: New Docking Methods, Expanded Force Field, and Python Bindings. J Chem Inf Model 2021, 61, 3891–3898, doi:10.1021/ACS.JCIM.1C00203/SUPPL_FILE/CI1C00203_SI_002.ZIP.
- Sarkar, A.; Concilio, S.; Sessa, L.; Marrafino, F.; Piotto, S. Advancements and Novel Approaches in Modified AutoDock Vina Algorithms for Enhanced Molecular Docking. Results Chem 2024, 7, 101319, doi:10.1016/J.RECHEM.2024.101319.
- Vanommeslaeghe, K.; Hatcher, E.; Acharya, C.; Kundu, S.; Zhong, S.; Shim, J.; Darian, E.; Guvench, O.; Lopes, P.; Vorobyov, I.; et al. CHARMM General Force Field (CGenFF): A Force Field for Drug-like Molecules Compatible with the CHARMM All-Atom Additive Biological Force Fields. J Comput Chem 2010, 31, 671, doi:10.1002/JCC.21367.
- Du, X.; Li, Y.; Xia, Y.L.; Ai, S.M.; Liang, J.; Sang, P.; Ji, X.L.; Liu, S.Q. Insights into Protein–Ligand Interactions: Mechanisms, Models, and Methods. International Journal of Molecular Sciences 2016, Vol. 17, Page 144 2016, 17, 144, doi:10.3390/IJMS17020144.
- Remorosa, A.G.B.; Tsai, P.W.; De Castro-Cruz, K.A.; Hsueh, C.C.; Chen, R.Y.; Chen, B.Y. Deciphering Characteristics of Macaranga Tanarius Leaves Extract with Electron Shuttle-Associated Anti-Inflammatory Activity via Microbial Fuel Cells, Molecular Docking, and Network Pharmacology. Biochem Eng J 2024, 208, 109345, doi:10.1016/J.BEJ.2024.109345.
- Liu, K.; Watanabe, E.; Kokubo, H. Exploring the Stability of Ligand Binding Modes to Proteins by Molecular Dynamics Simulations. J Comput Aided Mol Des 2017, 31, 201–211, doi:10.1007/S10822-016-0005-2/METRICS.
- Liu, K.; Kokubo, H. Exploring the Stability of Ligand Binding Modes to Proteins by Molecular Dynamics Simulations: A Cross-Docking Study. J Chem Inf Model 2017, 57, 2514–2522, doi:10.1021/ACS.JCIM.7B00412/SUPPL_FILE/CI7B00412_SI_001.PDF.

Reviewer 2 Report
Comments and Suggestions for Authors
The manuscript ‘Deciphering the regulatory potential of antioxidant and electron-shuttling bioactive compounds in Oolong tea’ presents an actual topic and the authors describe the conducted experiments quite comprehensively. However, it needs revision.
The main notes:
Methodology and Results.
The authors did not determine the luteolin content in the studied extracts chromatographically, but only determined the total flavonoid content. In this regard, why is the second half of the manuscript devoted specifically to luteolin? Please note that in section 2.2.2 the analysis is performed in terms of rutin!
Based on the data in Table 2, the content of catechins was quite high, especially in OTL-E, so it is not clear why the properties of this group of compounds were not discussed comprehensively. The authors should explain it!
Methodology.
2.1. Sample preparation. It is unclear why the authors used ethanol for extraction,n besides the water, and not its various dilutions with water?
Some wordings are unsuccessful, for example:
2.2.1. Total phenolic (TPC) – it should be 2.2.1. Total phenolic content (TPC)
2.2.2. Total flavonoid (TFC) – it should be 2.2.1. 2.2.2. Total flavonoid content (TFC)
2.2.3. Total condensed tannin (TCTC) – it should be 2.2.3. Total condensed tannin content (TCTC)
Please, indicate the wavelengths in the spectrophotometric analyses in this section - 2.3. Antioxidant activity
Main text and Abstract -
As for me, the manuscript contains some unjustified abbreviations as they occur only 1-2 times in the text – for instance, Smart Manager Software (SMS) in line 271. The other abbreviations are not deciphered, for instance, - OTL-E in the Abstract and in Tables 2, 3. In addition, it is worth removing abbreviations from the sections and subsections
The terms in vitro and in silico should be added to the keywords and the description of research methods and results.
Line 168 – why do the authors use water for dissolving all extracts (even if it was obtained with ethanol)?
Lines 430-437 - I recommend that the authors move this paragraph to the Discussion section.
I would recommend shortening the list of sources used a bit (at least by 10%), as it is a bit too long for an experimental article.
Conclusions
This formulation is not appropriate - While OTL-E demonstrated superior phytochemical and antioxidant properties …’
Round 2
Reviewer 1 Report
Comments and Suggestions for Authors
The authors have included the suggested points. However, they still need to complete some points.
Comment 1: To ensure the accuracy of docking poses, a comparison of X, Y, Z coordinates with Native structures' active sites is needed.
Comment 2: The provided figures in S1-S6 are single, separate figures. Authors need to provide 3D superimposed figures for complexes. All ligands should be visualized at the same binding pocket. Please see the examples:
https://www.mdpi.com/entropy/entropy-24-00593/article_deploy/html/images/entropy-24-00593-g001.png
https://www.mdpi.com/molecules/molecules-25-03589/article_deploy/html/images/molecules-25-03589-g002.png
Comment 3: To validate the docking binding affinity on which the authors claimed their findings should be recalculated using MMPBSA free energy calculation methods. Authors can choose a stable time frame for the run calculation in a short time. Moreover, to establish the convergence of MDS data, PCA and Free Energy landscape data is essential.
All new data should be discussed in the results and discussion sections.
The authors need to address the suggested points to strengthen the study.
Reviewer 2 Report
Comments and Suggestions for Authors
The authors substantially improved this manuscript, and now it could be recommended for the acceptance
Author Response
We sincerely thank you for your thoughtful comments and suggestions, which have greatly contributed to the improvement of our manuscript. We are truly grateful for your time, expertise, and encouraging feedback. Your positive recommendation means a great deal to us, and we are honored by your support.
Thank you once again for helping us enhance the quality of our work.